# The SUN1-SPDYA interaction plays an essential role in meiosis prophase I

Yanyan Chen[1,2,8], Yan Wang[1,2,8], Juan Chen[3,4], Wu Zuo[1,2], Yong Fan[3,4], Sijia Huang[3,4], Yongmei Liu[3,4], Guangming Chen[3,4,5], Qing Li [2,6], Jinsong Li [2,6], Jian Wu [3,4], Qian Bian [3,4], Chenhui Huang [3,4✉] & Ming Lei [3,4,7✉]

Chromosomes pair and synapse with their homologous partners to segregate correctly at the first meiotic division. Association of telomeres with the LINC (Linker of Nucleoskeleton and Cytoskeleton) complex composed of SUN1 and KASH5 enables telomere-led chromosome movements and telomere bouquet formation, facilitating precise pairwise alignment of homologs. Here, we identify a direct interaction between SUN1 and Speedy A (SPDYA) and determine the crystal structure of human SUN1-SPDYA-CDK2 ternary complex. Analysis of meiosis prophase I process in SPDYA-binding-deficient SUN1 mutant mice reveals that the SUN1-SPDYA interaction is required for the telomere-LINC complex connection and the assembly of a ring-shaped telomere supramolecular architecture at the nuclear envelope, which is critical for efficient homologous pairing and synapsis. Overall, our results provide structural insights into meiotic telomere structure that is essential for meiotic prophase I progression.

[1] State Key Laboratory of Molecular Biology, Shanghai Institute of Biochemistry and Cell Biology, Center for Excellence in Molecular Cell Science, Chinese Academy of Sciences, Shanghai, China. [2] University of Chinese Academy of Sciences, Beijing, China. [3] Ninth People's Hospital, Shanghai Jiao Tong University School of Medicine, Shanghai, China. [4] Shanghai Institute of Precision Medicine, Shanghai, China. [5] Laboratory of Vector Biology and Pathogen Control of Zhejiang Province, Huzhou University, Huzhou Central Hospital, Zhenjiang, China. [6] State Key Laboratory of Cell Biology, CAS Center for Excellence in Molecular Cell Science, Shanghai Institute of Biochemistry and Cell Biology, Chinese Academy of Sciences, Shanghai, China. [7] Key laboratory of Cell Differentiation and Apoptosis of Chinese Ministry of Education, Shanghai Jiao Tong University School of Medicine, Shanghai, China. [8] These authors contributed equally: Yanyan Chen, Yan Wang. ✉email: huangchh@shsmu.edu.cn; leim@shsmu.edu.cn

M eiosis produces haploid gametes through a single round of DNA replication followed by two successive nuclear divisions for homolog segregation and sister chromosome separation, respectively[1]. To allow faithful segregation of the homologs, homologous chromosomes undergo intimate pairing, recombination, and synapsis in meiotic prophase I[2]. The attachment of telomeres to the inner nuclear membrane (INM) and subsequent movement along the INM in prophase I is a conserved phenomenon in meiosis and is essential for homology search, recombination, and synapsis[3,4]. In mammals, the meiosis-specific LINC complex composed of KASH5 and SUN1 provides the binding site for telomeres for transmission of cytoskeletal forces to regulate telomere dynamics[5–9]. Knockout of either SUN1 or KASH5 in mice causes defects in homologous pairing and meiotic arrest, suggestive of an essential role of the meiotic LINC complex in prophase progression[8,9]. While it is clear that SUN1 defines the INM attachment sites for telomeres, the nature of this interaction and its regulation still remain unknown.

In addition to the LINC complex, another INM-associated complex consisting of meiosis-specific structural molecules TERB1, TERB2 (telomere repeat binding bouquet formation proteins 1 and 2), and MAJIN (membrane-anchored junction protein) was recently identified to be implicated in tethering telomeres to the INM by direct interaction with shelterin protein TRF1[10–14]. Disruption of the TERB1–TERB2–MAJIN (TTM) complex results in complete depletion of telomeres from the INM and abolishment of the association between telomeres and the LINC complex, indicating that the TTM-dependent telomere–INM attachment is essential for the telomere–LINC complex connection[13–15]. Besides the LINC and the TTM complexes, Speedy/RINGO (rapid inducer of G2/M progression in oocytes) A (SPDYA) and CDK2 (cyclin-dependent kinase 2) also have been shown to play important roles in the anchorage of telomeres to the INM during meiotic prophase I[16–18]. SPDYA serves as a meiosis-specific regulator of CDK2 and is indispensable for the recruitment of CDK2 to telomeres[16,17]. Knockout of either SPDYA or CDK2 results in the same meiotic defects, including accumulated nucleoplasmic telomeres and prophase arrest with aberrant synapsis, implying that SPDYA and CDK2 likely function together in regulating telomere–INM attachment, and are therefore critical for homolog pairing in early prophase I[16–19]. However, it is still unclear how SPDYA and CDK2 regulate telomere attachment to the INM and how this process is coupled to the LINC complex-regulated movement of telomeres at the INM.

Here, we identify a direct interaction between SUN1 and SPDYA, determine the crystal structure of the SUN1–SPDYA–CDK2 ternary complex, and provide in vivo evidence that SUN1–SPDYA-mediated LINC–telomere connection is essential for the assembly of a nuclear envelope (NE)-attached ring-shaped telomeric supramolecular complex that plays an essential role in meiotic prophase I progression.

## Results

### Identification of the interaction between SUN1 and SPDYA.

To understand how the LINC complex is connected to telomeres in meiosis, we employed yeast two-hybrid (Y2H) interaction analysis to systematically examine the potential interactions between SUN1 with telomeric shelterin proteins and meiotic factors that have been reported to play important roles at telomeres[13–18]. We discovered a direct interaction between the nucleoplasmic region of SUN1 (residues 1–210) and SPDYA (Fig. 1a). When ectopically co-expressed in U-2 OS cells, SUN1 could efficiently recruit SPDYA to the nuclear membrane (Supplementary Fig. 1a), confirming the direct interaction between SUN1 and SPDYA. Using this membrane recruitment assay, we

**Table 1 Crystal data collection and refinement statistics.**

| | SUN1$_{SBM}$-SPDYA$_{ERD}$-CDK2 |
|---|---|
| *Data collection* | |
| Wavelength (Å) | 0.91905 |
| Space group | C222$_1$ |
| Cell dimensions | |
| a, b, c (Å) | 59.3, 183.5, 103.8 |
| α, β, γ (°) | 90.0, 90.0, 90.0 |
| Resolution (Å) | 3.2 |
| $R_{merge}$ (%) | 10.0 (57.3)* |
| I / σI | 13.4 (2.0)* |
| Completeness (%) | 99.3 (94.3)* |
| Redundancy | 5.2 (4.0)* |
| *Refinement* | |
| Resolution (Å) | 42.59-3.20 |
| No. of reflections | 9095 |
| $R_{work}$/$R_{free}$ (%) | 25.2/30.0 |
| No. of atoms | |
| CDK2 | 2396 |
| SPDYA | 1216 |
| SUN1 | 166 |
| *B*-factors (Å$^2$) | |
| CDK2 | 81.5 |
| SPDYA | 75.9 |
| SUN1 | 93.4 |
| R.m.s. deviations | |
| Bond lengths (Å) | 0.008 |
| Bond angles (°) | 0.040 |
| Ramachandran plot (%) | 93.4 |
| Favored region | 100.0 |
| Allowed region | 0.0 |
| Outlier region | |

*Highest resolution shell is shown in parenthesis.

found that a fragment of SPDYA (residues 61–213) that includes the conserved Ringo domain (residues 68–200) was sufficient for interacting with a short segment of SUN1, SUN1$_{SBM}$ (residues 136–155, the SPDYA-binding motif) (Fig. 1b and Supplementary Fig. 1b–e). Notably, the short extension (residues 61–67) N-terminal to the Ringo domain of SPDYA is essential for the interaction with SUN1 (Supplementary Fig. 1b, c). Hereafter, we will refer to this SUN1-binding fragment (residues 61–213) as the extended Ringo domain of SPDYA (SPDYA$_{ERD}$) (Fig. 1b). On the other hand, when SUN1$_{SBM}$ was fused with membrane-bound MAJIN, it efficiently recruited SPDYA to the INM (Supplementary Fig. 1f), suggesting that SUN1$_{SBM}$ motif alone was capable of associating with SPDYA. Given that SPDYA also mediates interaction with shelterin protein TRF1[20], the identification and characterization of the direct association between SUN1 and SPDYA unveiled a key linkage in the interaction network from the LINC complex to the telomere in meiosis.

### Crystal structure of the SUN1$_{SBM}$–SPDYA$_{ERD}$–CDK2 ternary complex.

To understand how SUN1 is recognized by SPDYA, we purified the SUN1$_{SBM}$–SPDYA$_{ERD}$–CDK2 ternary complex and determined its crystal structure using a single-wavelength anomalous dispersion method at a resolution of 3.2 Å (Fig. 1c and Table 1). The calculated electron density map allowed us to unambiguously trace the entire complex, including the SUN1$_{SBM}$ polypeptide (Supplementary Fig. 2a). The structure reveals that the conformation of the SPDYA$_{ERD}$–CDK2 moiety in the ternary complex is almost identical to that of the previously reported crystal structure of the SPDYA$_{Ringo}$–CDK2 complex except for the N-terminal strand β1 in SPDYA$_{ERD}$, which is essential for SUN1$_{SBM}$ binding but absent in the binary crystal structure

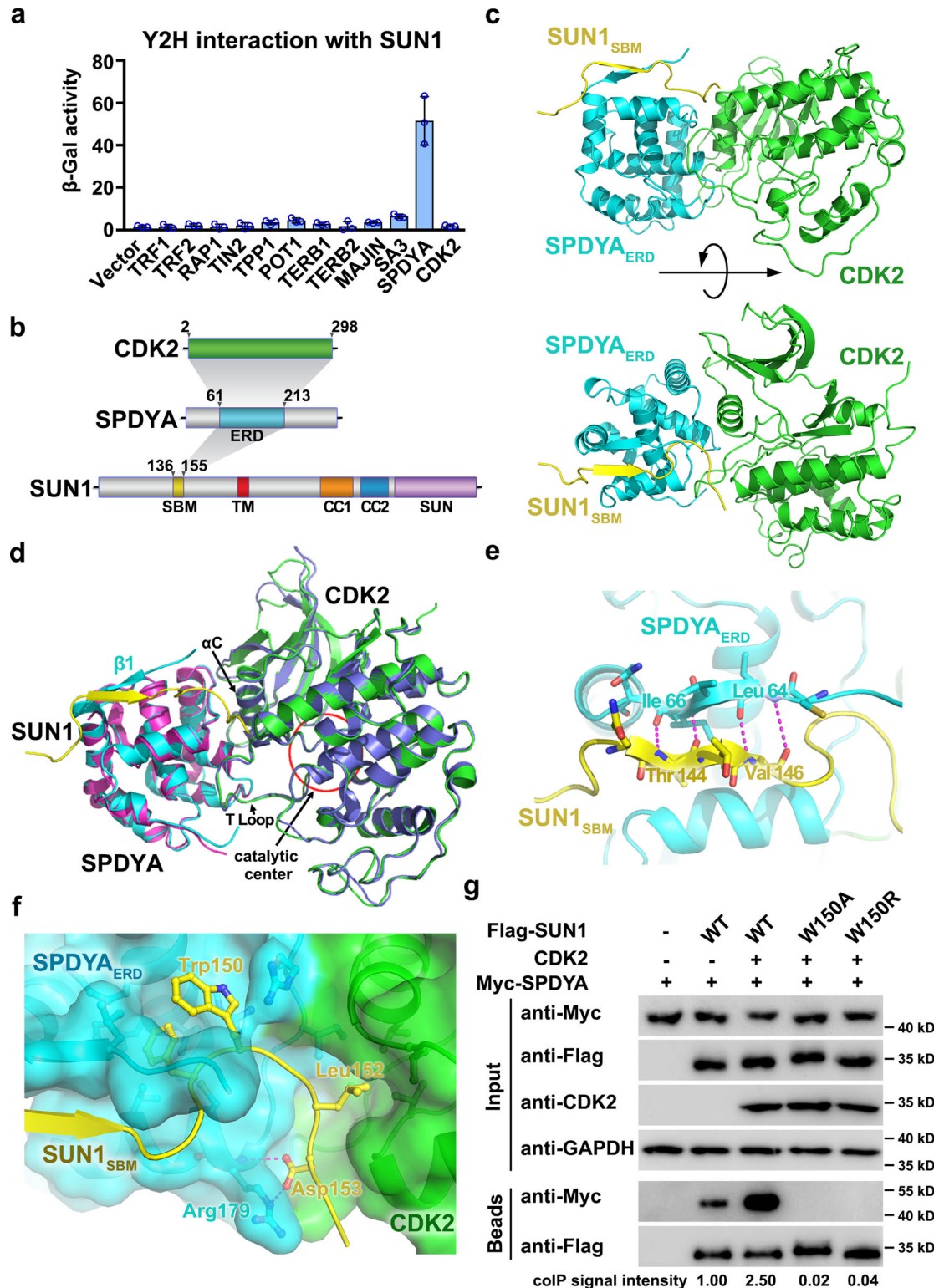

(Fig. 1d)[21]. In both structures, CDK2 is stabilized in an active configuration through an interface involving the activation loop (T-Loop) and the αC helix of CDK2 and one side of the SPDYA_{Ringo} domain (Fig. 1d)[21].

In the ternary complex structure, the SUN1_{SBM} polypeptide chain mostly contacts the surface of SPDYA_{ERD} on the opposite side of the catalytic center of CDK2 (Fig. 1d), suggesting that the binding of SUN1_{SBM} does not interfere with CDK2's catalytic activity. This is confirmed by an in vitro kinase assay, in which

the CDK2 activity is modulated by the presence of SPDYA but not the SUN1_{SBM} polypeptide (Supplementary Fig. 2b). Residues $_{143}$QTTV$_{146}$ of SUN1_{SBM} form an antiparallel β sheet with β1 of SPDYA_{ERD} (Fig. 1e). After the β strand, SUN1_{SBM} makes a sharp turn so that SUN1$^{Phe149}$ and SUN1$^{Trp150}$ snugly fit into a hydrophobic cavity under strand β1 of SPDYA_{ERD} (Fig. 1f and Supplementary Fig. 2c). At the C-terminus of SUN1_{SBM}, the sidechain of Leu152 points into a hydrophobic pocket between the small and large lobes of CDK2, indicating that CDK2 may

**Fig. 1 SUN1 directly interacts with SPDYA. a** Yeast two-hybrid (Y2H) interaction analysis of the interaction between N-terminus of SUN1 (residues: 1–210) and the indicated proteins. Interactions between LexA-SUN1 and N-terminally GAD-fused indicated proteins were analyzed by measuring the β-galactosidase activity produced by the reporter gene. Vector, vector only. Data are presented as the mean ± SD for $n = 3$ independent experiments. **b** Domain organization of SUN1, SPDYA, and CDK2. The shaded areas indicate the interactions among these proteins. The interacting domains are labeled and highlighted in different colors. ERD extended Ringo domain, SBM SPDYA-binding motif, TM transmembrane, CC coiled-coil domains. **c** Ribbon diagrams of two orthogonal views of the SUN1$_{SBM}$–SPDYA$_{ERD}$–CDK2 ternary complex. SUN1$_{SBM}$, SPDYA$_{ERD}$, and CDK2 are colored in yellow, blue, and green, respectively. **d** Superimposed structures of the SUN1$_{SBM}$–SPDYA$_{ERD}$–CDK2 ternary complex and the binary SPDYA$_{Ringo}$–CDK2 complex[21]. SUN1$_{SBM}$, SPDYA$_{ERD}$, and CDK2 of the ternary complex are displayed in yellow, cyan, and green, respectively. SPDYA$_{Ringo}$ and CDK2 of the binary complex are displayed in magenta and blue, respectively. **e** SUN1$_{SBM}$ forms an antiparallel β sheet with β1 of SPDYA$_{EXD}$. Residues important for the interaction are shown as stick models. **f** Detailed interactions at the interfaces between SUN1$_{SBM}$ and SPDYA$_{ERD}$. **g** Co-IP measurement of ectopically expressed human Flag-SUN1$_{1-210}$, Myc-SPDYA, and CDK2 in HEK293T cells. The immunoprecipitation is performed with anti-Flag beads. The levels of each protein in the input and IP samples were analyzed by immunoblotting with the indicated antibodies. Band intensity of blots was quantified by the ImageJ software (NIH, https://imagej.nih.gov/ij/) and the co-IP signal for the sample with expression of WT Flag-SUN1$_{1-210}$ and Myc-SPDYA was set as 1. Source data are provided as a Source Data file.

also play a role in the binding with SUN1 (Fig. 1f and Supplementary Fig. 2c).

To corroborate the structural analysis, we first examined whether CDK2 contributes to the interaction between SUN1 and SPDYA–CDK2. Co-immunoprecipitation (co-IP) experiments revealed that the presence of CDK2 indeed substantially enhanced the interaction between SUN1 and SPDYA (Fig. 1g). Next, we examined whether missense mutations of SUN1 at the interface could disrupt the interaction between SUN1 and SPDYA–CDK2. We focused on the SUN1$^{Trp150}$ residue and found that substitution of this residue with alanine or arginine completely abolished the interaction as revealed by co-IP analyses (Fig. 1g). This result was further confirmed by both membrane recruitment assay and surface plasmon resonance analysis (Supplementary Fig. 3a, b). The same conclusion was also obtained when the interactions between mouse Sun1 mutants (mutations at Trp151, equivalent to human SUN1$^{Trp150}$) and Spdya-Cdk2 were analyzed, consistent with the fact that the interface residues are highly conserved between human and mouse proteins (Supplementary Fig. 3c–e).

**The SUN1–SPDYA interaction is required for gametogenesis in mice.** In order to evaluate the in vivo function of the SUN1–SPDYA interaction during meiosis, we generated knock-in mice with a SPDYA-binding-deficient mutation W151R in the *Sun1* gene using the CRISPR-Cas9 method (Supplementary Fig. 4a). The heterozygous mice appeared healthy and fertile and were intercrossed to produce homozygous offspring. Heterozygous and homozygous mutant progeny were identified by PCR and DNA sequencing with genomic DNA and cDNA (Supplementary Fig. 4b, c). Quantitative real-time PCR (qPCR) and western blotting analyses of 14-day-old mouse testes revealed that the in vivo mRNA and protein expression levels of SUN1 are not altered by the mutation (Supplementary Fig. 4d, e).

Homozygous *Sun1$^{W151R/W151R}$* (hereafter referred to as *W151R*) mice appeared overtly normal compared to the wild-type (WT) ones. However, inspection of reproductive tissues unveiled severe defects in their gross development (Fig. 2a). Histological analyses revealed that *Sun1* mutant seminiferous tubules were narrower and devoid of spermatids and spermatozoa (Fig. 2b), and only abnormal spermatocyte-like cells were accumulated in some tubules (Supplementary Fig. 5a). TdT-mediated dUTP nick-end labeling (TUNEL) assay exhibited a high prevalence of apoptotic cells in the mutant tubules, indicating that spermatocytes were massively eliminated (Fig. 2c, d). Consistently, no mature sperms appeared in the cauda epididymal lumen of *W151R* male mice (Fig. 2b). Similarly, mutant female mice showed degeneration of ovaries that lacked growing follicles and mature oocytes (Fig. 2e). Further analysis by

fluorescence-activated cell sorting (FACS) revealed an arrest of spermatogenesis process at the tetraploid stage of mutant primary spermatocytes (Supplementary Fig. 5b). Immunofluorescence (IF) staining of SYCP1 and SYCP3, the structural components of the axial and lateral elements (AEs and LEs) and of the transverse filaments of the central element (CE) of the synaptonemal complex (SC)[22,23], confirmed that the most advanced spermatocytes and fetal oocytes in *W151R* mice were at a zygotene-pachytene (hereafter called pachytene-like) stage, in which the formation of autosomal SC was not complete (Fig. 2f, g and Supplementary Fig. 5c, d). Notably, although a subset of SYCP3 filaments displayed SYCP1 staining signal in *W151R* spermatocytes, super-resolution microscopy analysis by stimulated emission depletion (STED) revealed that the majority if not all of the chromosomal synapsis were aberrant (Supplementary Fig. 5e). To investigate the underlying reason for the observed synapsis defect, we next performed IF-FISH (fluorescence in situ hybridization) using chromosome painting probes to examine the homologous association in spermatocytes and observed that homologs were separated from each other in the majority of *W151R* pachytene-like spermatocytes (Fig. 2h, i and Supplementary Fig. 5f), suggesting that defective synapsis in *W151R* spermatocytes is due to inefficient homolog association.

We further assessed the spatial organization of meiotic chromosomes in *W151R* mutant spermatocytes at higher resolution by performing high-throughput chromosome conformation capture (Hi-C) analysis[24]. We compared the relationship between chromatin contact probability and genome separation in *W151R* spermatocytes versus several previously published WT zygotene and pachytene Hi-C datasets using the contact probability P(s) analysis (Supplementary Fig. 6a)[25,26]. In *W151R* spermatocytes, the P(s) curve of autosomes exhibits a power-law decay with a slope of ~0.6 between 100 kb and 1 Mb, similar to the P(s) shape observed previously in WT zygotene spermatocytes[25]. However, the P(s) curve of *W151R* mutant drops at a shorter genomic separation compared to WT zygotene spermatocytes, suggesting that profound changes in chromosome organization occurred in these abnormal spermatocytes. Recent studies have revealed that the topologically associating domains (TADs) are largely lost in the meiotic prophase, while the A/B compartment organization persists[25–28]. By performing eigenvector decomposition analyses, we showed that the compartment identity of genomic regions was largely unchanged in the *W151R* mutant (Fig. 3a and Supplementary Fig. 6b). However, the compartment strength in the *W151R* mutant was attenuated compared to that in WT zygotene spermatocytes (Supplementary Fig. 6c, d).

Compared to the moderate changes in the intrachromosome organization, the association among different chromosomes

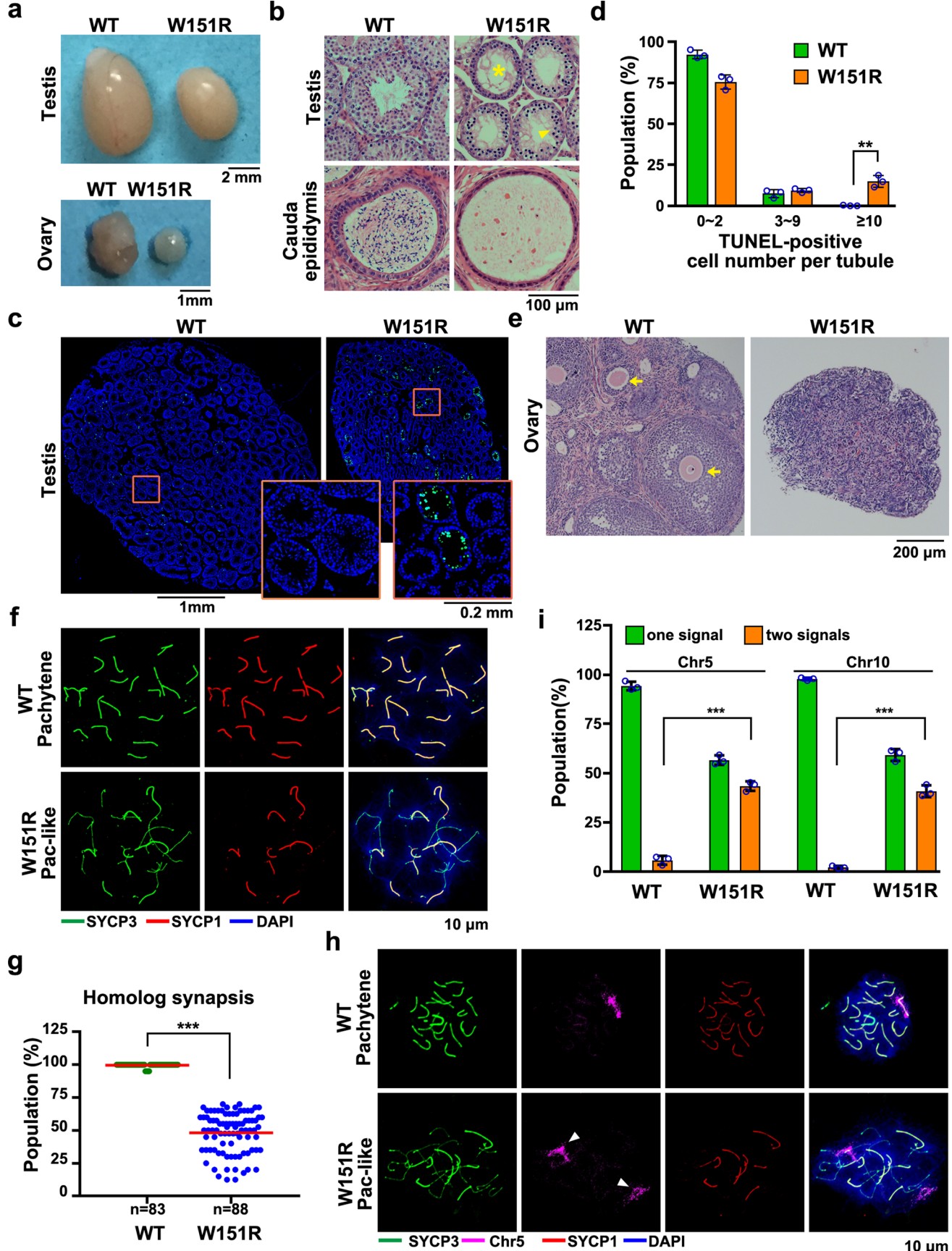

**Fig. 2 The SUN1–SPDYA interaction is required for gametogenesis in mice. a** Testes and ovaries from 6-week-old WT and *W151R* mouse littermates. Scale bars, 2 mm (for testis) and 1 mm (for ovary). **b** Hematoxylin and eosin-stained histological cross-sections of 6-week-old testes and epididymis. Arrowhead indicates abnormal spermatocyte-like cells in mutant seminiferous tubules. Asterisk shows vacuolated seminiferous lacking spermatids and spermatozoa. No sperm was observed in the epididymis of mutant mice. Scale bar, 100 μm. **c** TUNEL staining (green) of apoptotic cells in testis sections from 6-week-old mice. DNA was stained by DAPI (blue). Scale bars, 1 mm; 0.2 mm (for insets). **d** Population of TUNEL-positive tubules shown as the mean ± SD (*n* = 3 mice of each indicated genotype). A two-sided Student's *t* test was performed, **$P$ = 0.0021. **e** Ovary tissue sections from 6-week-old mouse littermates stained with hematoxylin and eosin. Arrowheads indicate oocytes and growing follicles in WT ovary. Scale bar, 200 μm. **f** IF staining of spermatocyte chromosome spreads for SYCP3 (green) and SYCP1 (red). (Pac-like: pachytene-like). Scale bar, 10 μm. **g** Population of fully paired homologous chromosomes in each WT pachytene or *W151R* pachytene-like spermatocytes from panel **f**. A total of *n* = 83 WT and *n* = 88 *W151R* spreads were counted. Red lines indicated the mean values. A two-sided Student's *t* test was performed, **$P$ = 2.59E-70. **h** IF staining of SYCP3 (green) and SYCP1 (red) and FISH detection of chromosome 5 (Chr5) on spermatocyte spreads from adult testes of different genotypes. Scale bar, 10 μm. **i** Population of spermatocyte spreads containing associated or separated chromosome 5 (Chr5) or chromosome 10 (Chr10) from panel **h** and Supplementary Fig. 5f. Data are presented as the mean ± SD (*n* = 3 mice of each indicated genotype). A two-sided Student's *t* test was performed, ***$P$ = 3.7E-5 (Chr5) and ***$P$ = 2.6E-5 (Chr10). For each mouse, >50 spreads were counted for quantification. Source data are provided as a Source Data file.

displayed more dramatic changes in the *W151R* mutant, highlighted by the loss of chromosome end congregation (Fig. 3b)[25]. Mouse chromosomes are acrocentric. In WT zygotene spermatocytes, the centromeric ends of chromosomes exhibited high frequencies of interchromosomal interactions (cen–cen), while interacting less frequently with the noncentromeric ends of other chromosomes (cen–tel) (Fig. 3b, c). Similarly, the noncentromeric chromosome ends also preferentially associated with each other (tel–tel), resulting in an X-shaped interchromosomal interaction pattern on Hi-C heatmaps (Fig. 3b, c). In *W151R* mutant, the interchromosomal interactions among the centromeric chromosome ends and among the noncentromeric chromosome ends decreased by ~3- and 2.5-fold, respectively (Fig. 3b, c), indicating a remarkable decrease in chromosome end contacts in mutant spermatocytes. Taken together, these results unveiled a key role of the SUN1–SPDYA interaction in promoting telomere clustering and chromosome alignment with correct topology, which is essential to homolog pairing, synapsis, and thereby the process of meiosis prophase I.

**The SUN1–SPDYA interaction is essential for programmed DSB repair and homologous recombination.** Repair of the programmed DNA double-strand breaks (DSBs) by the meiotic recombination machinery correlates with SC formation and homologous recombination[29]. The observation of defective synapsis prompted us to examine DSB processing in mutant spermatocytes. In WT spermatocytes, phosphorylated H2AX (γ-H2AX) initially appeared during leptotene when DSBs start to occur, then was distributed throughout the entire nucleus at the zygotene stage coupled with homologous chromosome pairing, and finally the signal of γ-H2AX was restricted solely within the XY body at pachytene after the completion of autosomal synapsis (Supplementary Fig. 7a)[29]. IF analysis revealed a WT-type pattern of γ-H2AX staining from leptotene through zygotene in mutant spermatocytes (Supplementary Fig. 7a), indicating that the initiation of meiotic DSB repair was not affected in mutant nuclei. However, we observed prolonged staining of γ-H2AX in *W151R* pachytene-like spermatocytes, suggestive of defective DSB repair and incomplete SC formation (Supplementary Fig. 7a).

To gain further information on the defect of DSB repair in mutant mice, we analyzed the distribution of DMC1, a protein marker for early repair of DSBs which assembles on the axial and lateral elements (AEs and LEs) of WT zygotene chromosomes and disappears later at the pachytene stage[30,31]. IF analysis showed that *W151R* pachytene-like spermatocytes displayed persistent DMC1 foci at asynapsed SYCP3 filaments Supplementary Fig. 7b, c), suggestive of a defect in homologous recombination. In mammalian meiotic prophase I, the early recombination intermediates containing DMC1 are gradually replaced by the

middle intermediates containing MSH4 and eventually by the late intermediates marked by MLH1[32,33]. IF Staining of *W151R* pachytene-like spermatocytes revealed an overall decrease in the number of MSH4 foci (Supplementary Fig. 7d, e), consistent with the observation of prolonged DMC1 staining (Supplementary Fig. 7b, c). Furthermore, no MLH1 foci appeared on *W151R* synapsed LEs (Supplementary Fig. 7f), suggestive of no crossover recombination. This was in accordance with the observation of meiotic arrest prior to pachytene in mutant spermatocytes. Taken together, we conclude that the SUN1–SPDYA interaction is essential for meiotic programmed DSB repair and homologous recombination.

**The SUN1–SPDYA interaction regulates the telomere–LINC complex connection and the telomere–INM attachment.** SUN1 provides the binding site for telomeres at the INM to transmit cytoskeletal forces for rapid meiotic prophase chromosome movements (RPMs)[3,34]. In WT pachytene spermatocytes, both SUN1 and KASH5 formed discrete foci at the tips of chromosomes (Fig. 4a and Supplementary Fig. 8a)[8,9]. Strikingly, neither SUN1 nor KASH5 appeared on the SYCP3 filaments in *W151R* mutant spermatocytes (Fig. 4a and Supplementary Fig. 8a), suggesting that the connection between telomeres and the meiotic LINC complex is completely abolished in the absence of the SUN1-SPDYA interaction. It is noteworthy that, compared to WT zygotene and pachytene spermatocytes that displayed clear telomere-bound SPDYA foci, mutant zygotene and pachytene-like spermatocytes exhibited reduced intensity of SPDYA staining at telomeres (Fig. 4b, c), indicating that the SUN1–SPDYA interaction also plays a role in the stability of the telomere association with SPDYA.

We next examined whether the telomere–INM attachment was affected in *W151R* mutant spermatocytes. IF-FISH revealed that about half of telomeres in mutant nuclei remained at the INM (Fig. 4d, e). Notably, the same phenomenon was also observed in *Sun1*[−/−] spermatocytes[13,15], indicating that the SUN1–SPDYA interaction plays a key role in telomere–INM connection. Many TTM complex foci appeared at the chromosome tips in mutant spreads and IF-FISH analysis revealed a complete co-localization of MAJIN with the INM-located telomeres in *W151R* spermatocytes (Supplementary Fig. 8b–f), suggesting that the residual telomere–INM attachment in *W151R* nuclei was likely mediated by the INM-anchored TTM complex. This result is consistent with our previous data that disruption of the TTM complex completely removes meiotic telomeres from the INM[15]. Nonetheless, given that substantial telomeres were detached from the INM in mutant nuclei (Fig. 4d, e), we concluded that the LINC complex plays an important role in the stability of the telomere–INM attachment.

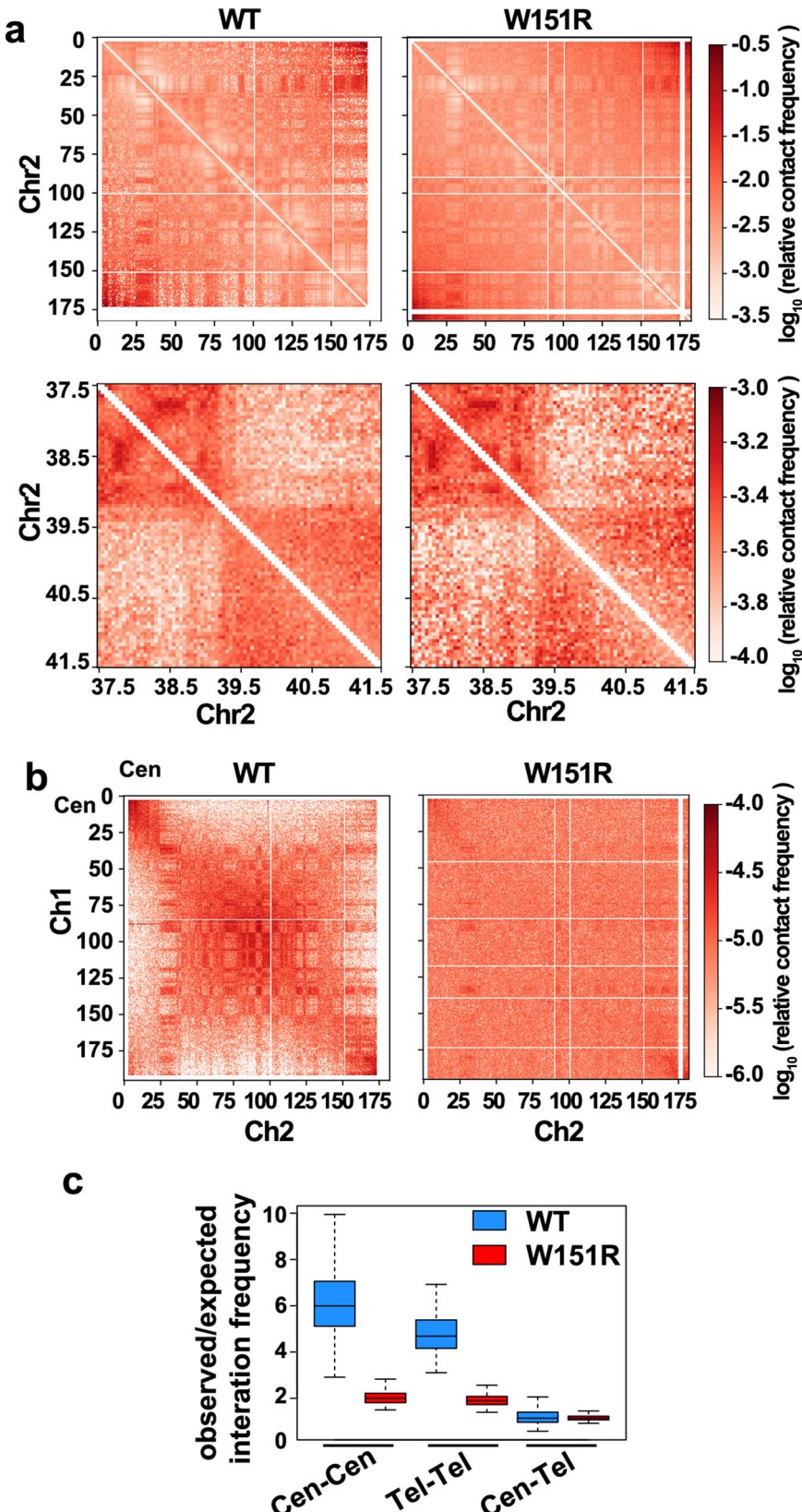

**Disruption of the SUN1–SPDYA interaction interferes with the distribution of the LINC complex and lamina.** The LINC complex drags telomeres for RPMs[3,34]. We set out to investigate whether disruption of the telomere–LINC complex connection interfered with the dynamics of the LINC complex on the nuclear envelope (NE) in mutant spermatocytes. In WT spermatocytes,

both components of the LINC complex, SUN1 and KASH5, exhibited discrete foci that overlapped with telomere signals at the nuclear periphery during most of the time in prophase I (Fig. 5a and Supplementary Fig. 9a). Consistent with previous studies[35], these SUN1 and KASH5 foci clustered at one side of the nuclear envelope in a transitory stage when telomere bouquets were

**Fig. 3 Disruption of the SUN1–SPDYA interaction impairs extensive alignment of chromosome ends. a** Heatmaps showing normalized Hi-C interactions (observed/expected) for the entire Chr2 (top, 50-kb bin) or a 4 Mb Chr2 region (bottom, 10-kb bin) in WT zygotene and *W151R* pachytene-like spermatocytes. WT zygotene data is from Patel et al.[25]. **b** Heatmaps showing normalized Hi-C interactions between Chr1 and Chr2 (50-kb bin) in WT zygotene and *W151R* pachytene-like spermatocytes. In WT spermatocytes, the chromosome ends preferentially interact with each other in trans. In *W151R* mutant, the clustering of chromosome ends is greatly diminished. Cen centromeric regions. **c** Boxplot shows changes in normalized interchromosomal interactions among the 10 Mb regions at the centromeric (Cen) or noncentromeric (Tel) ends of autosomes in *W151R* mutant ($N = 171, 171, 171, 171, 342,$ 342 for each column, respectively). Boxes, middle 50% of interchromosomal interaction scores. Center bars, lower bounds of boxes and upper bounds of boxes indicate median scores, the first quartile of scores ($Q\_1$) and the third quartile ($Q\_3$) of scores, respectively. Upper and lower whiskers correspond to $\min(\max(x), Q\_3 + 1.5 * IQR)$ and $\max(\min(x), Q\_1 - 1.5 * IQR)$. IQR interquartile range.

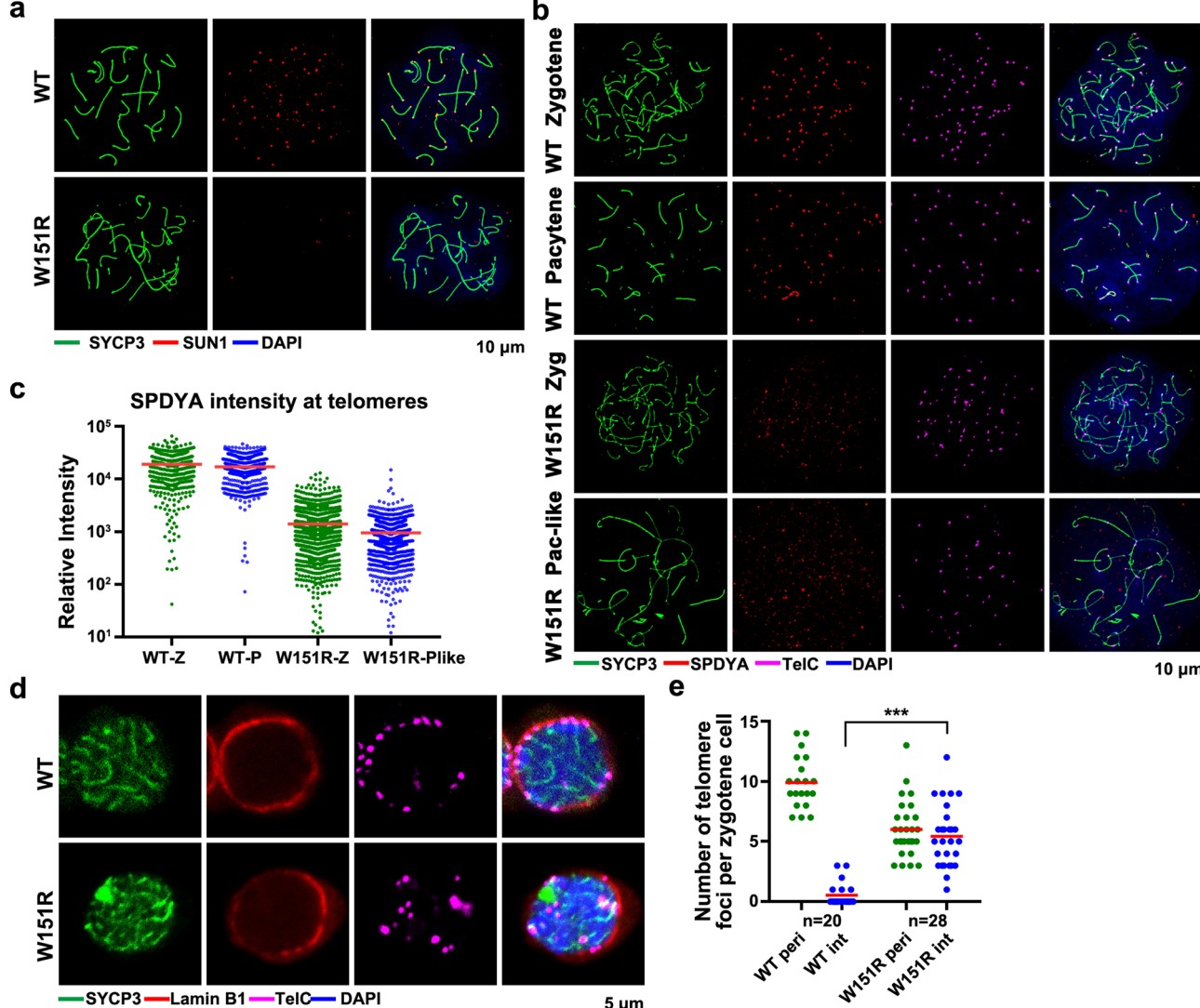

**Fig. 4 The SUN1–SPDYA interaction regulates telomere–LINC complex connection and telomere NE attachment. a** IF staining of spermatocyte chromosome spreads for SYCP3 (green) and SUN1 (red). DNA was stained by DAPI (blue). Scale bar, 10 μm. **b** IF-FISH staining of spermatocyte chromosome spreads for SYCP3 (green), SPDYA (red), and telomeric DNA (TelC, magenta). (Zyg zygotene, Pac-like pachytene-like). Scale bar, 10 μm. **c** Quantification of the relative intensity of SPDYA foci at telomeres in panel **b**. A total of $n = 8$ WT Zygotene spreads (557 telomeres) (WT-Z), $n = 12$ WT pachytene spreads (498 telomeres) (WT-P), $n = 19$ *W151R* zygotene spreads (1269 telomeres) (W151R-Z) and $n = 15$ *W151R* pachytene-like spreads (733 telomeres) (W151R-Plike) were analyzed. Red lines indicated the mean values. **d** Equator images of structurally preserved zygotene spermatocytes stained for SYCP3 (green), Lamin B1 (red), and telomere FISH (TelC, magenta). Scale bar, 5 μm. **e** Quantification of the number of telomere foci at the nuclear periphery (peri) or internal domain (int) as shown in panel **d**. A total of $n = 20$ WT and $n = 28$ *W151R* spermatocytes were counted. Red lines indicated the mean values. A two-sided Student's $t$ test was performed, ***$P = 2.6\text{E-}10$. Source data are provided as a Source Data file.

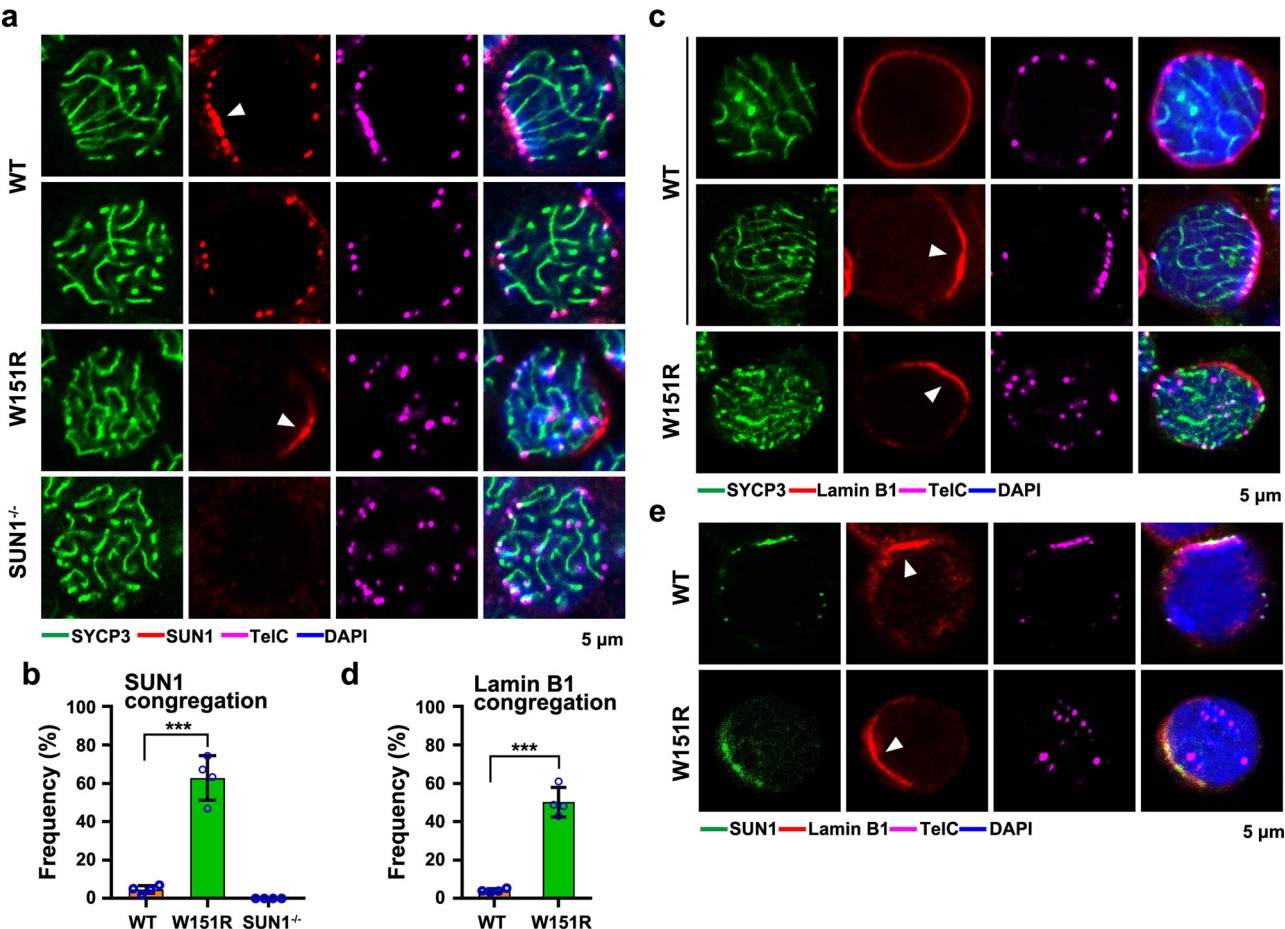

**Fig. 5 Disruption of the SUN1–SPDYA interaction interferes with the distribution of the LINC complex and lamina. a** Representative images of zygotene spermatocytes in testis sections stained for SYCP3 (green), SUN1 (red), and telomere FISH (TelC, magenta). Arrowheads indicate clustered signals for SUN1. DNA was stained by DAPI (blue). Scale bar, 5 μm. **b** Population of nuclei that display SUN1 congregation in panel **a**. Data are presented as the mean ± SD (*n* = 4 mice of each indicated genotype). A two-sided Student's *t* test was performed, ***P = 6.2E-5. For each experiment, >100 spermatocytes from each indicated genotype were counted. **c** Equator images of structurally preserved zygotene spermatocytes stained for SYCP3 (green), Lamin B1 (red), and telomere FISH (TelC, magenta). Arrowheads indicate congregated signals for Lamin B1. Scale bar, 5 μm. **d** Population of nuclei exhibiting Lamin B1 congregation in panel **c**. Data are presented as the mean ± SD (*n* = 4 mice of each indicated genotype). A two-sided Student's *t* test was performed, ***P = 2.2E-5. For each experiment, >100 spermatocytes from each indicated genotype were counted. **e** Equator images of structurally preserved zygotene spermatocytes stained for SUN1 (green), Lamin B1 (red), and telomere FISH (magenta). Arrowheads indicate congregated signals for Lamin B1. Scale bar, 5 μm. Source data are provided as a Source Data file.

formed (Fig. 5a and Supplementary Fig. 9a). In contrast, both SUN1 and KASH5 displayed a continuous patch distribution without co-localization with telomeres in more than half of *W151R* zygotene and pachytene-like nuclei (Fig. 5a, b and Supplementary Fig. 9a, b), a strikingly different configuration from the discrete foci observed in WT spermatocytes, indicating that the dynamic distribution of the LINC complex at the NE is regulated by the connection with telomeres. Furthermore, the accumulation of spermatocytes with congregated LINC complex in *W151R* testis suggests that the signal(s) that triggers the resolution of the LINC complex clustering after the bouquet stage is missing in the mutant nuclei (Fig. 5a, b and Supplementary Fig. 9a, b).

The nuclear lamina, a structural component of the NE, mediates extensive contacts with chromatin and INM proteins[36]. Given that SUN1 was polarized at the INM in *W151R* spermatocytes (Fig. 5a), we asked whether the structure of the nuclear lamina was similarly impaired by disruption of the telomere–LINC connection. IF staining with antibody against Lamin B1 showed that lamina displayed largely uniform distribution at the nuclear periphery in most WT zygotene and

pachytene spermatocytes (Fig. 5c). Notably, a small portion of WT spermatocytes exhibited a distinct congregation of Lamin B1 at one side of the nuclear periphery with clustered telomere foci embedded (Fig. 5c, d), unveiling a dynamics process of Lamin B1 during the bouquet stage of meiosis. Similar to the LINC complex, Lamin B1 also showed a congregated patch distribution in most *W151R* pachytene-like spermatocytes without co-localization with telomere foci (Fig. 5c, d), suggesting that the dynamics process of Lamin B1 was interfered with by the loss of the telomere–LINC complex connection. This notion was further confirmed by the observation that the Lamin B receptor (LBR) and the lamina-associated polypeptide 2 (LAP2), two lamina-associated integral membrane proteins involved in maintaining the structural organization of the NE[37–39], displayed the same distribution patterns as those of Lamin B1 in both WT and *W151R* spermatocytes (Supplementary Fig. 9c–e). Moreover, these congregated patches appeared together and faced the pericentrin-labeled centrosome in *W151R* spermatocytes (Supplementary Fig. 9f), suggestive of a clustering arrangement of lamina related to the process of bouquet formation. It is noteworthy that the congregated Lamin B1 patches overlapped

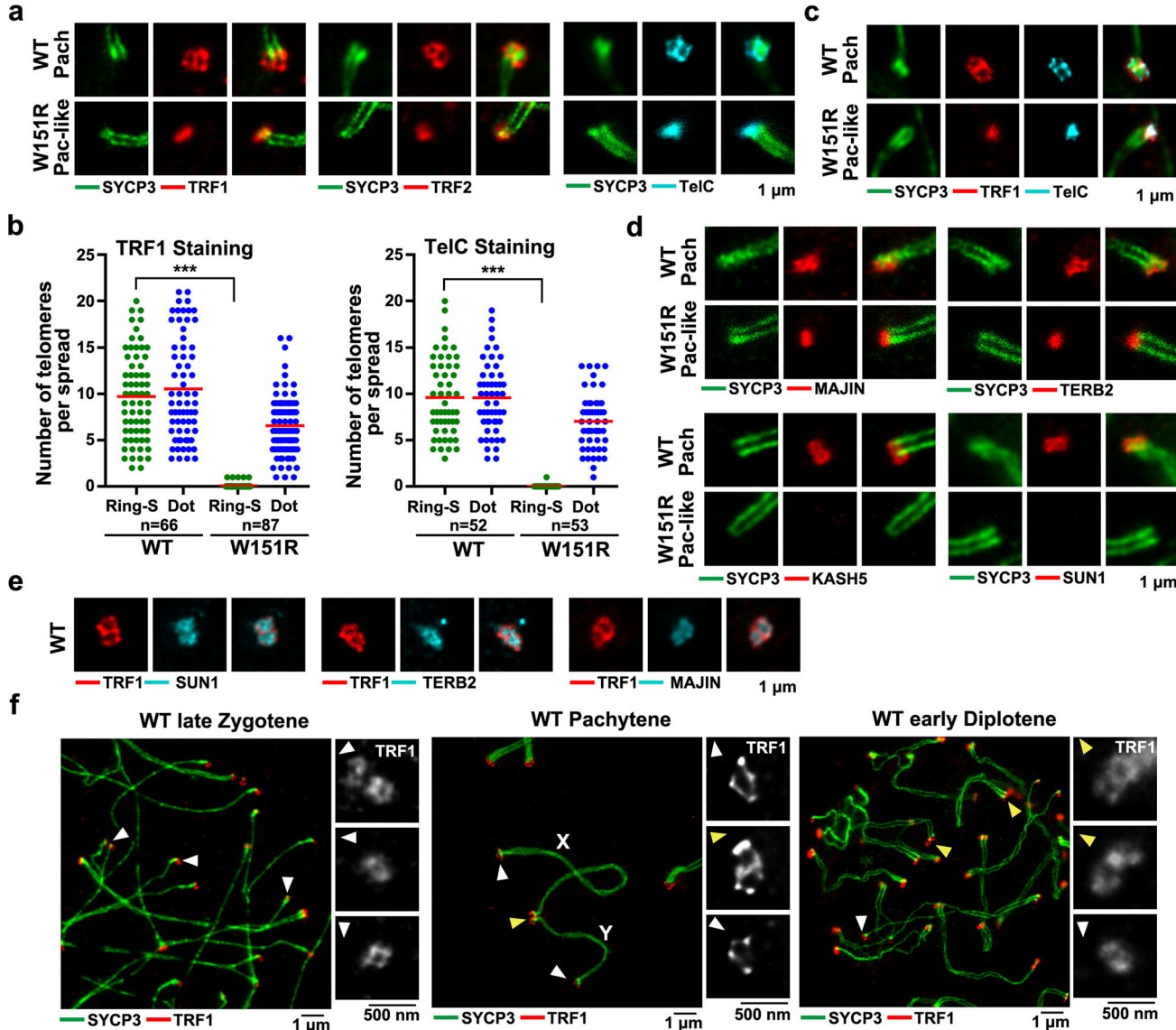

**Fig. 6 The SUN1–SPDYA interaction is required for telomere structures on the NE. a** IF analysis by STED of spermatocyte spreads stained for SYCP3 (green) and TRF1 (red), TRF2 (red), or telomeric DNA (telC, cyan). Scale bar, 1 μm. (Pach pachytene). **b** Quantification of the number of ring and figure-eight-like TRF1 and TelC structures per spread STED image from panel **a**. In each spermatocyte spread, only a subset of telomeres can be imaged. Ring-S, ring or figure-eight-like structures. Dot, focal structures. A total of $n = 66$ WT (for TRF1), $n = 87$ *W151R* (for TRF1), $n = 52$ WT (for TelC), and $n = 53$ *W151R* (for TelC) spermatocyte spreads were counted. Red lines indicated the mean values. A two-sided Student's $t$ test was performed, ***$P = 7.3E-42$ (TRF1) and ***$P = 8.9E-31$ (TelC). **c** IF-FISH analysis by STED of spermatocyte spreads stained for SYCP3 (green), TRF1 (red), and telomeric DNA (telC, cyan). Scale bar, 1 μm. **d** IF analysis by STED of spermatocyte spreads stained for SYCP3 (green) and MAJIN (red), TERB2 (red), KASH5 (red), SUN1 (red). Scale bar, 1 μm. **e** IF analysis by STED of spermatocyte spreads stained for TRF1 (red) together with SUN1 (cyan), TERB2 (cyan) or MAJIN (cyan). Scale bar, 1 μm. **f** IF analysis by STED of WT chromosome spreads stained for SYCP3 (green) and TRF1 (red). Insets show TRF1 ring structures at asynapsed telomeres (white arrowheads) and figure-eight-like structures at synapsed telomeres (yellow arrowheads), respectively. X and Y indicate chromosomes X and Y, respectively. Scale bars, 1 μm; 500 nm (insets). Source data are provided as a Source Data file.

with the SUN1 aggregates in both WT and *W151R* nuclei (Fig. 5e), suggesting that the dynamic process of lamina at the INM is likely correlated with the LINC complex. Further IF analysis showed that *Sun1*$^{-/-}$ spermatocytes exhibited the same congregated distribution of KASH5, Lamin B1, LAP2, and LBR as in *W151R* spermatocytes, indicating that the LINC complex is not required for lamina congregation at the INM (Supplementary Fig. 9g). Although the mechanism for lamina dynamics is still not clear, the dispersion of congregated lamina being affected in both *W151R* and *Sun1*$^{-/-}$ spermatocytes suggests that loss of the LINC–telomere connection interferes with lamina dynamics during meiotic prophase. Taken together, these data suggest that

the LINC complex and the lamina are likely regulated by coordinated but independent dynamic processes on the INM during meiotic prophase I and that the telomere–LINC complex connection plays an important role in these processes.

**The SUN1–SPDYA interaction is required for a ring-shaped telomere structure at the NE.** Next, we asked whether telomere structure at the INM was also interfered with by the loss of the telomere–LINC connection. To address this issue, we employed STED super-resolution microscopy to examine the telomere architecture. Remarkably, we observed ring-shaped and figure-eight-like structures of shelterin proteins TRF1 and TRF2 and

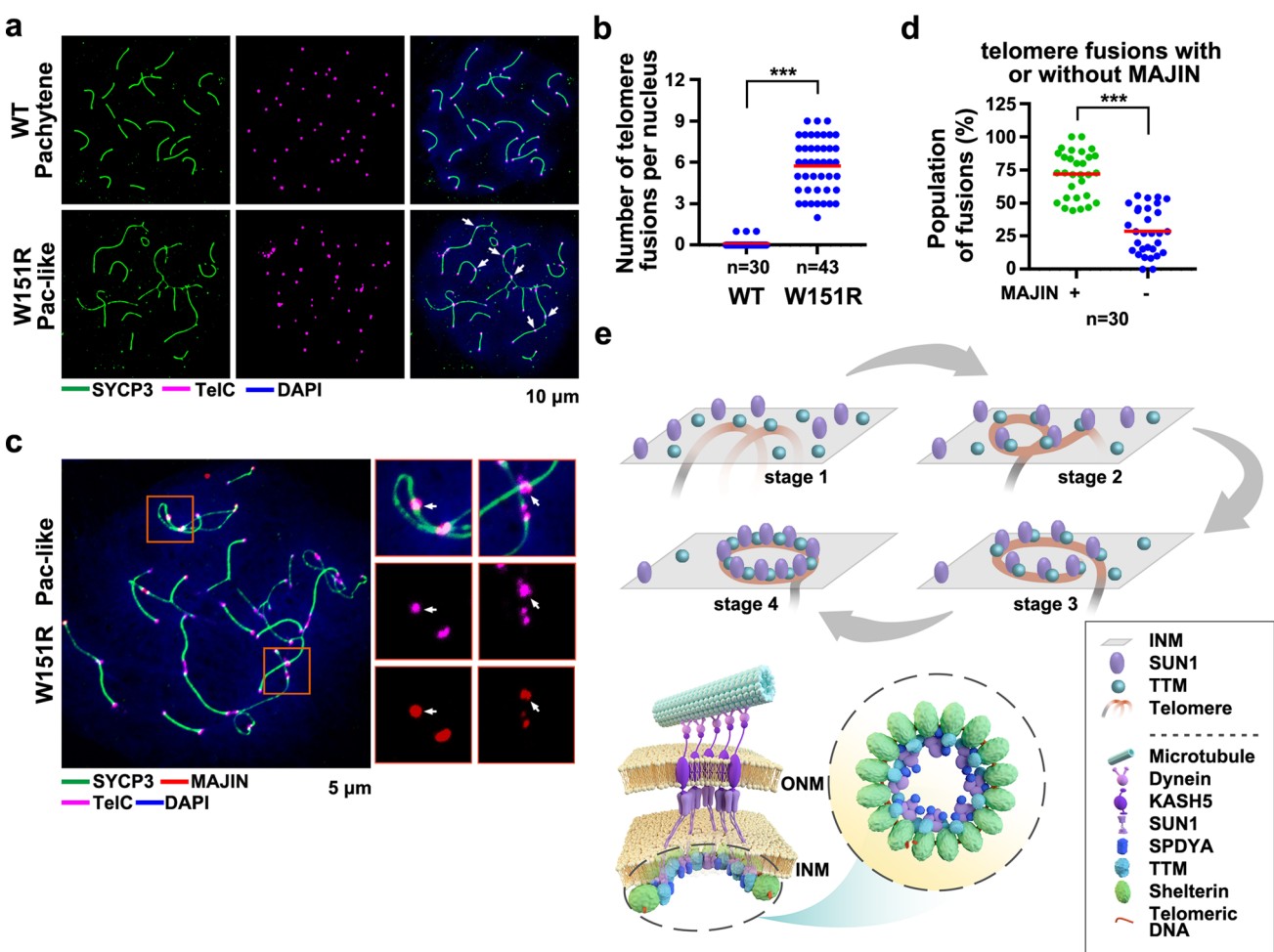

**Fig. 7 Loss of the SUN1–SPDYA interaction disrupts telomere integrity. a** IF-FISH images of chromosome spreads stained for SYCP3 (green) and telomeric DNA (telC, magenta). Arrowheads indicate telomere fusions and bridges. DNA was stained by DAPI (blue). **b** Quantification of the number of telomere fusions and bridges per nucleus of WT ($n = 30$) and *W151R* ($n = 43$) spermatocytes in panel **a**. Red lines indicated the mean values. A two-sided Student's *t* test was performed, \*\*\**P* = 1.9E-24. **c** IF-FISH analysis of spermatocyte spreads stained for SYCP3 (green), MAJIN (red), and telomeric DNA (telC, magenta). Insets are enlarged figures of the framed regions containing telomere fusions. **d** Quantification of frequency of telomere fusions and bridges with (+) or without (−) MAJIN staining signal in *W151R* ($n = 30$) spermatocytes from panel **c**. Red lines indicated the mean values. A two-sided Student's *t* test was performed, \*\*\**P* = 1.4E-13. **e** Schematic model of the assembly of the telomere supramolecular structure at the INM. The TTM complex mediates the initial attachment of telomeres to the INM (stage 1). Association of SUN1 with the INM-anchored telomere-TTM complex stabilizes the telomere–INM connection (stage 2). Dynein-driven telomere movements allow telomeres to encounter more TTM and LINC complexes (stage 3), till the formation of the ring-shaped architecture of the supramolecular complex (stage 4), in which the TTM, the LINC, and the SPDYA–CDK2 complexes are confined into the telomere rings (bottom). Source data are provided as a Source Data file.

telomeric DNAs at the tips of the LEs on WT pachytene chromosome spreads (Fig. 6a, b and Supplementary Fig. 10a, b). In contrast, no ring-shaped or figure-eight-like structures of TRF1, TRF2, or telomeric DNAs were observed in *W151R* pachytene-like spermatocytes (Fig. 6a, b and Supplementary Fig. 10a, b), suggesting that the formation of these unique meiotic telomere structures on the INM depends on the LINC–telomere connection mediated by the SUN1–SPDYA interaction. Notably, our IF-FISH data revealed a persistent co-localization of TRF1 and telomeric DNAs at chromosome ends in WT spermatocyte spreads (Fig. 6c and Supplementary Fig. 10c), arguing against the "cap exchange" model in which the shelterin complex was proposed to be displaced from telomeric DNAs at WT pachytene stage[13].

In addition to shelterin proteins and telomeric DNAs, components of the TTM and the LINC complexes also exhibited ring-shaped arrangement at chromosome ends in WT pachytene spreads (Fig. 6d), consistent with observations in a previous

study[8]. STED imaging clearly showed that the signals of SUN1, MAJIN, and TERB2 were all well confined in the ring-shaped telomere structure in WT spreads (Fig. 6e), implying that both the TTM and the LINC complexes very likely assemble together with telomeres into a ring-shaped supramolecular telomeric architecture at the INM. Consistent with the notion that the telomere–LINC complex connection was disrupted in mutant spermatocytes (Fig. 4a and Supplementary Fig. 8a), there were no signals of the LINC complex at telomeres in *W151R* spreads (Fig. 6d). But surprisingly, no ring-shaped structure of the TTM complex was observed in *W151R* pachytene-like spermatocytes (Fig. 6d and Supplementary Fig. 10d), even though the TTM–telomere connection can still maintain some telomere attachment to the INM in the absence of the SUN1–SPDYA interaction (Supplementary Fig. 8b–f). These observations suggest that both the TTM-mediated telomere–INM attachment and the telomere–LINC connection are essential for the formation of meiotic ring-shaped telomere structures at the INM.

The ring-shaped telomeric architecture was evident in WT spermatocytes from late zygonema to early diplonema (Fig. 6f). In particular, in WT late zygotene spermatocytes, ring-shaped TRF1 signals appeared at the tips of asynapsed AEs (Fig. 6f, left), suggesting that the formation of the telomeric ring-shaped structure is independent of homolog synapsis. Similarly, many asynapsed termini of sex chromosomes in WT pachytene spermatocytes also exhibited ring-shaped structures, while the synapsed ends appeared as figure-eight-like arrangement (Fig. 6f, middle), indicating that each figure-eight-like architecture at the tips of SCs consists of two adjacent but separated ring-shaped structures of telomeres of homolog chromosomes. Consistent with this idea, desynapsed telomeres in WT diplotene spermatocytes only exhibited ring-shaped structures (Fig. 6f, right). On the other hand, the SYCP3 signal was excluded from the telomere rings (Fig. 6a), suggesting that telomeres, even those from homolog chromosomes, are spatially separated and not implicated in synapsis. In aggregate, we conclude that the SUN1–SPDYA interaction is essential for the assembly of the telomeric ring-shaped architecture at the INM, which is independent of homolog synapsis.

**Loss of the SUN1–SPDYA interaction leads to telomere fusions at the INM.** To investigate the potential function of the observed telomere rings, we then examined the telomere integrity in *W151R* spermatocytes that lacked the ring-shaped telomere architecture at the INM. IF-FISH analysis revealed a marked increase in the frequency of telomere fusions in *W151R* pachytene-like spreads (Fig. 7a, b). In somatic cells, shelterin protein TRF2 is essential for telomere capping and loss of TRF2 results in massive chromosome fusion at telomeres[40,41]. Notably, TRF2 signals were largely intact at fused telomeres in *W151R* pachytene-like spreads (Supplementary Fig. 11a). In addition, we found no persistent γ-H2AX signals at the fused telomeric ends (Supplementary Fig. 11b), suggestive of the absence of telomere dysfunction-induced foci (TIFs) at the fusion sites. These data indicated that telomere fusions in *W151R* spermatocytes are unlikely caused by the loss of TRF2. Short-term treatment with okadaic acid, which stimulates an artificial chromatin transition from meiotic prophase to metaphase I[42], induced desynapsis and homolog separation in *W151R* pachytene-like spermatocytes without distinguishable changes in the frequency of telomere fusions (Supplementary Fig. 11c, d). This result demonstrated that the observed telomere fusions in mutant pachytene-like nuclei are not due to protein factors that hold homolog chromosomes together during SC formation in WT spermatocytes. Notably, IF-FISH analysis revealed that the majority of the fused telomeres (~75%) on *W151R* spreads exhibited strong MAJIN signals (Fig. 7c, d), suggesting that formation of the telomere fusion due to the loss of the telomere–LINC connection likely occurs at the INM in mutant spermatocytes. Consistent with this idea, we observed very few telomere fusions in TERB2-deficient spermatocytes, where telomeres were completely depleted from the INM because of the disruption of the TTM complex (Supplementary Fig. 11e, f)[15]. Taken together, these data suggest that the SUN1–SPDYA interaction and thus the ring-shaped supramolecular complex plays an important role in telomere protection that prevents fusions of chromosome ends at the INM.

## Discussion

How telomere is attached to the INM during meiosis prophase I is a central question in regeneration biology and is important for gamogenesis and human fertility. The meiosis-specific LINC complex plays a key role in this process through the assembly of a stable telomeric supramolecular architecture together with the TTM and the shelterin complexes at the NE[8,9,13,14,17]. However, despite extensive studies, the lack of structural information of the interaction between the telomere and the LINC complex has hindered in-depth understanding of this complicated process.

Our structural and functional data reported here, when combined with previous studies, propose an integrated model for the assembly of a ring-shaped telomere supramolecular complex at the INM in meiotic prophase I. First, interaction between TERB1 of the TTM complex and TRF1 in the telomere-associated shelterin complex initiates the connection of telomeres with the INM in leptotene spermatocytes (Fig. 7e, stage 1)[12,14]. At this stage, most telomeres adopt linear conformations and likely not all the telomere-associated shelterin complexes bind to the TTM complexes (Fig. 7e, stage 1). Loss of this TTM-mediated telomere–INM interaction completely prevents telomere attachment to the INM[13,15]. Next, the LINC complex binds to the TTM-tethered telomeres promoted by the direct interaction between SUN1 and SPDYA to achieve a more stable association of telomeres with the INM (Fig. 7e, stage 2). Then, dynein-dependent movement of telomeres at the INM transduced through the LINC complex allows telomeres to encounter more TTM and LINC complexes. Finally, all the telomere-associated shelterin complexes are saturated by both the TTM and LINC complexes so that the entire telomere region is tightly attached to the INM to form in a ring-shaped supramolecular architecture (Fig. 7e, stages 3 and 4). It is likely that the LINC complex is a key organizer for this supramolecular arrangement. Disruption of the SUN1–SPDYA interaction by the *Sun1^{W151R}* mutation abolishes the telomere–LINC complex connection and causes a complete loss of telomeric ring-sharp architecture (Figs. 4a and 6a), resulting in partial telomere detachment from the INM and reduced stability of SPDYA at telomeres (Fig. 4b–e). Further studies will be needed to fully understand how the LINC complex facilitates the formation of the telomeric ring-sharp architecture at the INM.

What are the functions of the ring-shaped arrangement of the telomere supramolecular complex in meiosis? First, this unique structure could mediate highly concentrated interactions between shelterin and NE-anchored TTM and LINC complexes and thus maintain a stable telomeres-INM association and facilitate the force transmission from the LINC complex to telomeres during dynein-dependent RPMs that is essential for chromosome homology search (Fig. 7e)[43,44]. In addition, another advantage of the ring-shaped structure over linear confirmation is that the ring-shaped structure could integrate all the mechanical forces to drive the telomere movement at the INM in a more harmonic and efficient manner. Third, telomeres appear to require a different protection mechanism at the INM than in the nucleus, and the ring-shaped architecture is a meiotic specific protective structure to prevent telomeres from end-to-end fusions at the INM (Fig. 7a). Loss of this protective structure in *W151R* spermatocytes results in fusions of INM-attached telomeres (Fig. 7a, b). Furthermore, the ring-shaped telomere architecture might have another protective role in preventing unwanted entanglement of telomeres, either of homologs or of non-homologs, which could lead to misalignment of chromosomes and eventually caused aberrant SC formation (Supplementary Fig. 5e). In fact, the existence of the figure-of-eight structures strongly suggests that telomeres even from homologous chromosomes are spatially separated and are not synapsed by the SC complex, consistent with the observation that SYCP3 signals were excluded from the ring-shaped telomere structures revealed by the IF analysis (Fig. 6a).

CDK2 is a serine/threonine protein kinase that plays an important role in mitotic cell cycle regulation[45]. Deletion of CDK2 leads to meiotic arrest at a pachytene-like stage, suggesting

that CDK2 also plays an essential role in meiosis[18,19]. In meiotic prophase I, CDK2 is found at telomeres on the INM from leptonema toward diplotene and in the XY bodies at the pachytene stage[46]. Our data reveal that CDK2 can enhance the SUN1–SPDYA interaction and facilitate the formation of the meiosis-specific telomere–LINC supramolecular complex. Whether CDK2's kinase activity plays a direct role in the meiotic prophase is still not clear and awaits future investigations.

In summary, our work delineates a key linkage in the interaction network from the LINC complex to the telomere, reveals an essential role of the telomere–LINC connection in the spatial organization of meiotic chromosomes, and unveils a ring-shaped telomere supramolecular structure at the INM that is essential for meiotic prophase progression. Further structural studies of this telomere ring-shaped supramolecular structure by in situ three-dimensional cryo-electron microscopic tomography will be required to fully understand this important and sophisticated molecular architecture in meiosis.

## Methods

**Animals**. Adult WT C57BL/6J mice and ICR(CD-1) mice were purchased from Shanghai SLAC laboratory animal Co. Ltd. Mice were housed under constant ambient temperature ($22 \pm 2\,^\circ$C) and humidity ($55 \pm 10\%$), with an alternating 12 h light/dark cycle. Water and food were available ad libitum. Experimental protocols were approved by the regional ethical committee of the National Center for Protein Science Shanghai (approval #SIBCB-S342-1510-043). Every effort was made to minimize and refine the experiments to avoid animal suffering.

**Cell culture**. HEK293T and U-2 OS cells were purchased from the Type Culture Collection of the Chinese Academy of Sciences, Shanghai, China. Both cells were cultured in DMEM supplemented with 10% (v/v) fetal bovine serum (FBS) (Hyclone) at $37\,^\circ$C under 5% $CO_2$.

**Antibodies**. The following antibodies were used in this study: secondary antibodies for IF and western blotting: Alexa Fluor 647 Goat anti-Rabbit IgG (Invitrogen, A-21245, 1:500, RRID: AB_2535813), Alexa Fluor 647 Goat anti-Mouse IgG (Invitrogen, A-21236, 1:500, RRID: AB_2535805), Alexa Fluor 555 Goat anti-Rabbit IgG (Invitrogen, A-21429, 1:500, RRID: AB_2535850), Alexa Fluor 555 Goat anti-Mouse IgG (Invitrogen, A-21424, 1:500, RRID: AB_141780), Alexa Fluor 488 Goat anti-Rabbit IgG (Invitrogen, A-11008, 1:500, RRID: AB_143165), Alexa Fluor 488 Goat anti-Mouse IgG (Invitrogen, A-11001, 1:500, RRID: AB_2534069), DyLight 594 Goat anti-Mouse IgG (Invitrogen, 35511, 1:500, RRID: AB_1965950), DyLight 594 Goat anti-Rabbit IgG (Invitrogen, 35561, 1:500, RRID: AB_1965951), HRP-conjugated Goat anti-Mouse IgG (Proteintech, SA00001-1, 1:4000, RRID: AB_2722565), and HRP-conjugated Goat anti-Rabbit IgG (Proteintech, SA00001-2, 1:4000, RRID: AB_2722564). Rabbit antibodies against SYCP1 (Abcam, ab15087, 1:2000, RRID: AB_301633), SYCP3 (Abcam, ab15093, 1:2000, RRID: AB_301639), SOX9 (Millipore, ab5535, 1:500, RRID: AB_2239761), MSH4 (Abcam, ab58556, 1:500, RRID: AB_2770446), phos-Histone H2AX (Ser139) (Millipore, 05-636, 1:1000, RRID: AB_309864), Lamin B1 (Abcam, ab16048, 1:1000, RRID: AB_443298), c-Myc (Santa Cruz, sc-789, 1:500, RRID: AB_631274), TERB1 (1:1000)[12], TERB2 (1:2000)[15], MAJIN (1:2000)[15], SUN1 (1:1000)[15], and SPDYA (1:3000)[15]. Mouse antibodies against Phospho-Thr-Pro (CST, 9391, 1:2000, RRID: AB_331801), MLH1 (BD, 550838, 1:500, RRID: AB_2636290), SYCP3 (Abcam, ab97672, 1:2000, RRID: AB_10678841), c-Myc (Santa cruz, sc-40, 1:500, RRID: AB_627268), Flag (Sigma, F3165, 1:2000, RRID: AB_259529), actin (Sigma, A2228, 1:2000, RRID: AB_476697), GAPDH (Proteintech, 60004-1, 1:2000, RRID: ab_2107436), DMC1 (Abcam, ab11054, 1:500, RRID: AB_297706), Lamin B1 (Proteintech, 66095-1, 1:1000, RRID: AB_11232208), CDK2 (Santa cruz, sc-6248, 1:200, RRID: AB_627238), Pericentrin (Abcam, ab4448, 1:100, RRID: AB_304461), LBR (Abcam, ab122919, 1:500, RRID: AB_10902156), LAP2 (Abcam, ab185718, 1:500), KASH5 (this study, 1:2000), TRF1 (this study, 1:3000) and TRF2 (this study, 1:3000).

**Antibody production**. To generate KASH5 antibody, cDNA fragment-encoding mouse KASH5$_{421-612}$ was cloned into a modified pET28a (Novagen) vector with SUMO protein fused at the N-terminus after the His$_6$ tag and purified with Ni-NTA agarose beads. The purified recombinant protein was used to immunize BALB/C mice (Shanghai SLAC laboratory animal Co. Ltd) coupled with immunologic adjuvant QuickAntibody (Beijing Biodragon, KX0210042). Similarly, for the generation of TRF1 and TRF2 antibodies, sequences for mouse TRF1$_{49-255}$ and TRF2$_{86-290}$ were respectively cloned into pGEX6P-1 (GE Healthcare) vectors for expression as HRV3C-cleavable GST fusion proteins. The recombinant proteins were purified with glutathione Sepharose resin (GE Healthcare) and the purified

proteins were used to immunize mice. The obtained serums were used for IF assays.

**Protein expression and purification**. Sequences for human SUN1$_{131-160}$, SPDYA$_{61-213}$, and CDK2 were cloned into pGEX6P-1 (GE Healthcare), pET28a (Novagen) and pETDuet (Novagen) vectors for expression as GST-, His$_6$-SUMO-fusion proteins and non-tagged protein, respectively. A list of primers used for cloning is provided in Supplementary Data 1. Constructs were expressed separately in E. coli BL21 CodonPlus (DE3) cells (Stratagene). Transformed cells were grown at $37\,^\circ$C, cooled at $4\,^\circ$C for 30 min when OD reached 0.6, and protein expression was induced for 19 h with 0.3 mM IPTG at $18\,^\circ$C. Cells were harvested by centrifugation, resuspended in lysis buffer (25 mM HEPES, pH 7.5, 500 mM NaCl, 10% glycerol, 1 mM PMSF, 5 mM benzamidine, 1 µg/mL leupeptin, and 1 µg/mL pepstatin). The resuspended cells were mixed and lysed by sonication. The cell lysate was ultracentrifuged, and the supernatant was incubated with Ni-NTA agarose beads (Qiagen) at $4\,^\circ$C with rocking for 1 h. After extensive washing, the bound proteins were eluted using elution buffer (25 mM HEPES, pH 7.5, 300 mM NaCl, and 300 mM imidazole). The eluate which had been added ULP1 protease for cleaving His-SUMO tag was then bound to glutathione Sepharose resin and washed with wash buffer. The GST tag was then cleaved on-resin overnight with GST-3C protease, and the GST-3C and GST tag were separated from the SUN1$_{131-160}$–SPDYA$_{61-213}$–CDK2 complex with glutathione Sepharose. The protein sample was further purified by Superdex 200 column equilibrated with the buffer (25 mM HEPES, pH 7.5, 150 mM NaCl). The SUN1$_{131-160}$–SPDYA$_{61-213}$–CDK2 complex was concentrated to 6.2 mg/mL and stored at $-80\,^\circ$C.

For kinase assay, cDNA fragments of human full-length CDK2, SUN1$_{131-160}$, SPDYA$_{ERD}$, and FoxM1$_{526-748}$ were respectively cloned into pGEX6P-1, pGEX6P-1, pET28as, and pET28as vectors, and expressed separately in E. coli BL21 CodonPlus (DE3) cells. Recombinant GST-CDK2, GST-SUN1$_{131-160}$, and His-SUMO-FoxM1$_{526-748}$ proteins were purified using GST-tag or His-tag affinity chromatography. The SPDYA$_{61-213}$–CDK2 complex was purified from the mixed lysates of GST-CDK2 and His-SUMO-SPDYA$_{61-213}$ by a two-step (His-tag and GST-tag) affinity-purification scheme. His-SUMO-FoxM1$_{526-748}$ was prepared without the cleavage of the tag. For other proteins, GST and His-SUMO tags were cleaved. Protein samples were further subjected to Superdex 75 gel-filtration chromatography and fractions containing target proteins were concentrated and stored at $-80\,^\circ$C before use in the kinase assay.

**Crystallization and structure determination**. Crystallization of the SUN1$_{131-160}$–SPDYA$_{61-213}$–CDK2 complex was screened by sitting-drop vapor diffusion at $18\,^\circ$C. The crystals were grown in 0.15 M Ammonium citrate tribasic, pH 7.0, 12% PEG3350 for 1 day, and then gradually transferred into a harvesting solution (0.15 M Ammonium citrate tribasic, pH 7.0, 20% PEG3,350, 25% (v/v) glycerol) followed by flash-freezing in liquid nitrogen for storage. Datasets were collected under cryogenic conditions (100 K) at the Shanghai Synchrotron Radiation Facility (SSRF) beamlines BL19U1. A 3.2-Å dataset of the SUN1$_{131-160}$–SPDYA$_{61-213}$–CDK2 complex was collected at the wavelength of 0.91905 Å. The ternary structure was determined by molecular replacement using previously published SPDYA$_{Ringo}$–CDK2 complex structure (PDB: 7E34) as the searching model. The initial model was rebuilt with Coot[47], and the SUN1$_{SBM}$ polypeptide was then built into the electron density map. In the final Ramachandran plot, the favored and allowed residues are 93.4% and 100.0%, respectively. All the crystal structural figures were generated using the PyMOL Molecular Graphics System, Version 1.86 Schrödinger, LLC (http://www.pymol.org).

**Kinase assay**. Kinase reactions were performed in a volume of 20 µL in a buffer containing 25 mM HEPES, pH 7.0, 200 mM NaCl, 10 mM $MgCl_2$, 1 mM DTT, and 200 µM ATP. Substrate FoxM1 was diluted into the reaction buffer with a final concentration of 1 µM. Reactions were performed at $30\,^\circ$C for 20 min and quenched by the addition of SDS-PAGE loading buffer. Samples were boiled at $95\,^\circ$C for 5 min and separated on SDS-PAGE followed by immunoblotting using anti-phospho-Thr-Pro antibody (CST).

**Surface plasmon resonance (SPR)**. SPR technology-based binding assays were performed using a Biacore 8 K instrument (GE Healthcare) with running buffer PBS-P + (20 mM phosphate buffer, 2.7 mM KCl, 137 mM NaCl, 0.05% (v/v) surfactant P20, 2% DMSO) at $25\,^\circ$C. The recombinant SPDYA$_{ERD}$–CDK2 complex was covalently immobilized onto sensor CM5 chips by a standard amine-coupling procedure in 10 mM sodium acetate (pH 5.0). Purified SUN1$_{131-160}$ protein was serially diluted and injected onto a sensor chip at a flow rate of 30 µl/min for 120 s (contact phase), followed by 180 s of buffer flow (dissociation phase). The $K_D$ value was derived using Biacore 8 K Evaluation Software Version 1.0 (GE Healthcare) and steady-state analysis of data at equilibrium.

**Co-IP and western blotting**. cDNA fragments were constructed into pEGFP-N1 (Clontech) vector and transfected into HEK293T cells. After 36 h, cells were washed three times by cold PBS (Phosphate Buffered Saline) buffer, collected and lysed in lysis buffer (25 mM Tris-HCl, pH 7.4, 150 mM NaCl, 10% glycerol, 0.5% Triton X-100, and 1 mM EDTA, supplemented with EDTA-free complete protease

inhibitor (Roche)). Cell lysates were centrifuged (21,000×$g$, 15 min, 4 °C) and supernatants were precleared with Protein A Sepharose (GE Healthcare) and immunoprecipitated with anti-Flag beads (M2 Affinity Gel, Sigma) at 4 °C with rocking overnight. Precipitates were then washed with cold PBS and resuspended in SDS-loading buffer, boiled at 95 °C for 5 min, and separated immediately on SDS-PAGE for immunoblotting. To detect WT and mutant SUN1 proteins in spermatocytes, testis samples from 2-week-old mice were homogenized in PBS buffer. Cells were pelleted and lysed by sonication in lysis buffer. Cell lysates were centrifuged and supernatants were subjected to SDS-PAGE separation. For immunoblotting analysis, proteins were transferred to PVDF membranes (Millipore). The blots were incubated in blocking buffer (5% fat-free milk in PBS buffer supplemented with 0.05% TWEEN-20) at room temperature (RT) for 30 min and incubated with primary antibodies in blocking buffer at 4 °C overnight. Blots were then washed and incubated in the HRP-labeled secondary antibodies at RT for 1 h. After washing, blots were developed with ECL Prime Western Blotting System (GE Healthcare, RPN2232).

**Yeast two-hybrid interaction assay**. Yeast two-hybrid interaction assay was performed as described previously[48]. Briefly, human SUN1$_{1-210}$ was cloned into a pBTM116 vector (Clontech) with a LexA domain fused at the N-terminus. Candidate interacting proteins were cloned into a pACT2 vector (Clontech) with a GAL4 activation domain fused at the N-terminus. Constructs were co-transformed into the L40 yeast strain and yeast colonies were selected on -Leu -Trp plates. Yeast colonies were grown in defined complete (YC) media lacking Leu and Trp at 30 °C till the OD$_{600}$ reached 1.5–2. Cells were pelleted, resuspended in a lysis buffer (60 mM Na$_2$HPO$_4$, 40 mM NaH$_2$PO$_4$, 10 mM KCl, 1 mM MgSO$_4$, pH 7.0), and divided into two parts for OD$_{600}$ and β-galactosidase activity measurement, respectively. The β-galactosidase activities were measured by the OPNG (2-nitrophenyl β-D-galactopyranoside) assay[49]. Briefly, cell lysis was achieved by passing the cell samples through several freeze-thaw cycles with liquid nitrogen. The cell lysates were centrifuged and supernatants were added with 0.6 mg/ml OPNG and incubated at 30 °C for an optimized period of time followed by the addition of 0.3 M Na$_2$CO$_3$ to stop the reactions. β-galactosidase activity was measured at OD$_{420}$ and normalized to a total cell content OD$_{600}$.

**Nuclear membrane localization assay**. cDNA fragments of human SUN1 and MAJIN$^{DB}$ (the DNA-binding mutant of MAJIN[13]) were constructed into a pLVX-IRES-Puro vector (Clontech) with a GFP tag at the C-terminus. For the SUN1$_{136-155}$–MAJIN$^{DB}$ construct, cDNA fragment-encoding human SUN1$_{136-155}$ was inserted in the frame at the N-terminus of MAJIN$^{DB}$ with a GGSGGS link. Similarly, cDNA fragments of human SPDYA were constructed into a pLVX-IRES-Puro vector with a Myc tag at its N-terminus. Constructs were transfected into HEK293T cells with the help of plasmids pCMV-dR8.9 and pCMV-VSV-G with X-tremeGENE HP DNA Transfection Reagent (Roche) according to the manual. The lentiviral supernatants were harvested 36 h later, cleaned by 0.22-μm filter (Millipore), and used for co-infection of U-2 OS cells as indicated. After 36 h, infected U-2 OS cells were subjected to IF analysis.

**Generation and analysis of *Sun1* mutant mice**. *Sun1* mutant mice were generated by CRISPR/Cas9-mediated genome engineering[50]. A guide RNA cassette with sequence (*mSun1*-gRNA, Supplementary Data 1) was cloned into pX260 vector[51]. In vitro-transcribed Cas9 mRNA (100 ng/μL) and guide RNA (50 ng/μL), and donor oligo (100 ng/μL, *mSun1-W151R*-donor, Supplementary Data 1) were microinjected into the cytoplasm of fertilized eggs collected from C57BL/6J. The injected zygotes were cultured in M2 medium (Merck/Millipore) at 37 °C under 5% CO$_2$ overnight. The embryos that had reached the two-cell stage of development were implanted into the oviducts of pseudo-pregnant ICR foster mothers. The mice born from the foster mothers were genotyped by PCR of tail biopsy (primers: *mSun1*-GDPCR-3F and *mSun1*-GDPCR-3R, Supplementary Data 1). Founder mice were crossed to C57BL/6J mice to obtain germline transmission, and then heterozygous animals were backcrossed at least three generations to C57BL/6 J. Experimental homologous mice were generated by crossing heterozygous animals, and genotyping was performed by PCR on genomic DNA (primers: *mSun1*-GDPCR-3F and *mSun1*-GDPCR-3R, Supplementary Data 1) and testis cDNA (primers: *mSun1*_qPCR_F and *mSun1*-qPCR_R, Supplementary Data 1).

**Quantitative PCR**. Testes from 14-dpp (days past partum) mice were used to isolate total RNA using Trizol reagent (Invitrogen, 15596026). The reverse transcriptase reaction was immediately performed to amplify cDNA with 5 μg of RNA using the PrimeScript II 1st Stand cDNA Synthesis Kit (Takara, 6210A). Quantitative real-time PCR (qPCR) was performed with the Power SYBR Green PCR Master Mix (ABI, 4367659) using the ViiA 7 System (Thermo Fisher Scientific). *Sun1* expression was detected by qPCR using the following primers: *mSun1*_qPCR_F and *mSun1*_qPCR_R (Supplementary Data 1). Control *Gapdh* expression was detected using the following primers: *mGapdh*-qPCR_F and *mGapdh*_qPCR_R (Supplementary Data 1).

**Histological analysis**. Testes and ovaries were dissected and respectively fixed in Bouin's solution (100 mL Bouin's solution: 75 mL saturated picric acid Buffer, 25

mL 40% paraformaldehyde, and 5 mL glacial acetic acid) and 4% paraformaldehyde in PBS at RT overnight. Tissues were dehydrated through an ethanol series (10 min each in 50%, 75%, 95%, 100%), cleared twice in xylene, and embedded in paraffin. Tissue sections (5 μm) were prepared with a Leica RM2235 rotary microtome. Sections were stained with hematoxylin and eosin (H&E) (BBI, E607318). TUNEL assay was carried out with the ApoGreen Detection Kit (Yeasen, 40307ES20).

**PI staining for flow cytometry**. Testis cells were prepared for FACS analysis based on previous studies with minor modifications[52,53]. Briefly, each decapsulated testis was incubated in 5 ml DMEM medium supplemented with 1 mg/mL Collagenase type IV (Sigma) and 10 μg/mL DNase I (Roche) in a tube at 33 °C for 30 min. Then 1 mg/mL trypsin (Gibco) was added and samples were incubated for another 10 min. The dissociated testis samples were pipetted with a plastic disposable Pasteur pipet several times, followed by the addition of 10% FBS to inactivate trypsin. Testis cell samples were centrifuged, and pellets were resuspended in 50 μL PBS followed by the addition of 950 μL 70% (v/v) ethanol. Cells were fixed at 4 °C overnight, washed once with PBS, resuspended in PI staining solution (PBS supplemented with 0.1% Triton X-100, 200 μg/mL DNase-free RNase and 20 μg/mL PI), incubated at 37 °C for 30 min and subjected to FACS analysis with a BD FACS Calibur Flow Cytometer. Data obtained from flow cytometry were analyzed using FlowJo software (Tree Star).

**Hi-C and data analysis**. Pachytene-like spermatocytes were isolated from 1 to 2-month-old SUN1 *W151R* mutant mice by Fluorescence-Activated Cell Sorting[52]. In situ Hi-C was performed on two independently prepared biological replicates using a previously published protocol[54]. Read mapping, as well as filtering and iterative correction, were performed using scripts derived from those at https://bitbucket.org/mirnylab/hiclib. The mapped Hi-C fragments of replicates were combined and converted into a cooler format using the cooler package[55] (https://github.com/open2c/cooler). Heatmap generation, calculation of expected interaction frequency, eigenvector decomposition, compartment strength analysis, as well as comparison of chromosome ends associations between genotypes were performed on combined interaction matrices using the cooltools package (https://github.com/open2c/cooltools).

**Squash and spread preparations**. Squashed and surface-spread spermatocytes were prepared as previously described with minor modifications[56,57]. Briefly, for squashed spermatocytes, freshly extracted 14-dpp testes were minced into small pieces with tweezers and fixed in 2% PFA and 0.05% Triton X-100 for 10 min. Small pieces of tubules were then put on a slide rinsed with ethanol and chloroform mixture (1:1), covered with a coverslip, and gently pressed to distribute the material. The slide was immersed in liquid nitrogen, and the coverslip was gently removed. The slide was washed in PBS and used immediately for immunostaining. For spermatocyte chromosome spreads, testes were dissected, and seminiferous tubules were separated and placed in a hypotonic extraction buffer (30 mM Tris-HCl, pH 8.2, 50 mM sucrose, 17 mM sodium citrate, 5 mM EDTA, 0.5 mM DTT, and protease inhibitor cocktail (Roche)) for 10 min. Tubules were then moved to 100 mM sucrose solution (pH 8.2) and spermatocytes were detached by pipetting. Drops of cell suspension were then placed on the upper right corner of coverslips (Thermo, 12-545-84) soaked in a 1% paraformaldehyde solution (Sigma, P6148), pH 9.2, supplemented with 0.15% Triton X-100. Coverslips were dried overnight at RT and ready for immunostaining. Chromosome spreads of oocytes were prepared with 19-dpc (days post coitum) ovaries in the same way as for spermatocyte spreads.

**Immunofluorescence staining, telomere FISH, chromosome painting, and microscopy imaging**. For IF staining of cultured cells, cells were grown on coverslips (Thermo, T_7011254584), fixed by 4% paraformaldehyde, permeabilized with 0.15% Triton X-100, blocked with 5% BSA in PBS at RT for 1 h, and incubated with primary antibodies at 4 °C overnight. For IF staining of spermatocyte spreads, spreads were washed with PBS supplemented with 0.15% Triton X-100, blocked with 5% BSA in PBS at RT for 1 h, and incubated with primary antibodies at 4 °C overnight. For IF staining of paraffin-embedded tissue sections, sections on coverslips were dewaxed twice in fresh xylene for 15 min each, rehydrated in an ethanol series (10 min each in 100%, 95%, 75%, 50%, and 0% ethanol in distilled water). Epitope retrieval was carried out by boiling sections in TUF TARGET UNMASKING FLUID (Invitrogen, Z00R.0000) for 20 min. Sections were then washed with PBS supplemented with 0.5% Triton X-100 and blocked with 5% BSA in PBS at RT for 3 h, followed by incubation with primary antibodies at 4 °C overnight. Coverslips with cells, sections, or spreads were then washed and incubated with fluorescence-conjugated secondary antibodies at RT for 1 h. Coverslips were thoroughly washed in PBS supplemented with 0.1% Triton X-100, washed once in PBS supplemented with 1 μg/mL DAPI, air-dried, and subjected to microscopy imaging.

IF-FISH was performed as previously described with minor modifications[58]. Briefly, after incubation with secondary antibodies, coverslips were washed and fixed with 4% paraformaldehyde for 10 min, dehydrated through an ethanol series (2 min each in 75%, 85%, 100%) and air-dried. Samples were denatured at 85 °C for 5 min in the presence of 50 nM Cy3-or Cy5-labeled (CCCTAA)$_4$ PNA probes

(TelC) (Panagene) or XMP mouse chromosome 5 or 10 painting probes (Metasystems) in hybridization buffer (20 mM Tris-HCl, pH 7.5, 70% formamide and 1% blocking reagent (Roche, T7091700)). Hybridization was then carried out at 37 °C for more than 12 h. Coverslips were washed once with hybridization buffer, twice with 2×SSC (saline sodium citrate) (30 mM sodium citrate, pH 7.0, 0.3 M NaCl) and once with PBS supplemented with 1 µg/mL DAPI. Coverslips were then air-dried and subjected to microscopy imaging by Zeiss LSM 880 or Leica STED TCS SP8 using 63×NA/1.40 oil.

**Statistics and reproducibility**. For Figs. 2c, d, h, i, 5a–d and Supplementary Figs. 5f, 9a–e, experiments were repeated at least three times with different mouse littermates as independent biological replicates. Representative micrographs are shown and quantification results are presented as mean ± SD with a two-sided Student's *t* test performed. For Figs. 2f, g, 4b–e, 6a, b, 7a–d and Supplementary Figs. 5c, d, 8e, f, 11c–f, experiments were repeated three times with similar results using different mouse littermates and representative results are shown. A number of spermatocyte or ovary spreads, spermatocyte nuclei, tubules or telomeres were counted for quantification with a two-sided Student's *t* test performed. For Fig. 1a, yeast two-hybrid experiment was repeated independently three times and data are presented as the mean ± SD. For Fig. 1g and Supplementary Figs. 1a, c, e, f, 2b, 3a, d, e, data are representative of three independent experiments with similar results. For Supplementary Fig. 4d, qPCR experiment was repeated independently three times with testis cell samples of different mouse littermates, and data are presented as the mean ± SD. For Fig. 2a and Supplementary Figs. 4e and 5b. data are representative of three independent experiments with similar results using different mouse littermates. For Figs. 2b, e, 4a, 5e, 6c–f and Supplementary Figs. 5a, e, 7, 8a–d, 9f, g, 10, 11a, b, micrographs are representative of a minimum of ten images taken from at least three biological replicates with different mouse littermates showing similar results.

**Reporting summary**. Further information on research design is available in the Nature Research Reporting Summary linked to this article.

## Data availability
The crystal structure of the SUN1$_{SBM}$–SPDYA$_{ERD}$–CDK2 complex was deposited in PDB with accession code 7E34. The Hi-C data (Sun1 *W151R* pachytene-like spermatocytes) was deposited in the Gene Expression Omnibus database with accession code GSE155142. Other data supporting the findings of this study are available from the corresponding author on reasonable request. Source data are provided with this paper.

## Code availability
Codes used for Hi-C analysis in this manuscript are modified from the scripts available at https://github.com/open2c/cooltools/tree/master/docs/notebooks_old.

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

## Acknowledgements

We thank the staff members at BL18U1 and BL19U1 of the National Facility for Protein Science in Shanghai (NFPS), China for help with diffraction data collection and processing. We thank the staff members of the Large-scale Protein Preparation System , Integrated Laser Microscopy System and Animal Facility at the National Facility for Protein Science in Shanghai (NFPS), China for providing technical support and assistance in data collection and analysis. We thank S. He, S. Li, and H. Lu from Shanghai Jiao Tong University School of Medicine (SHSMU) for help with confocal microscopy and FACS. This work was supported by grants from the National Natural Science Foundation of China (U1632267 and 31525007 to M.L., 31971137 to C.H., 31801056 and 31970585 to Q.B.), the National Key Research and Development Program of China (2018YFC2000102 to M.L.), the Outstanding Academic Leader Program of Science and Technology Commission of Shanghai Municipality (16XD1405000 to M.L. and C.H.), the Incentive Project of High-level Innovation Team for Shanghai Jiao Tong University School of Medicine.

## Author contributions

Y.C., Y.W., C.H., and M.L. designed the study. Y.C., Y.W., and C.H. carried out biochemical assays. Y.C. crystallized the SUN1$_{SBM}$-SPDYA$_{ERD}$-CDK2 complex. J.W. and Y.C. collected diffraction data and determined the crystal structure. Y.C., J.C., Q.L., and J.L. constructed the knock-in mice. Y.C., Y.W., Y.L., Y.F., S.H., and C.H. performed the experiments to characterize the meiotic phenotype. W.Z., G.C., and Q.B. performed Hi-C analysis. Y.C., C.H., and L.M. wrote the manuscript. All authors were involved in data interpretation and writing of the manuscript.

## Competing interests

The authors declare no competing interests.
