## [Peer Review File · Nature Communications]

REVIEWER COMMENTS

Reviewer #1 (Remarks to the Author):

In this manuscript, Chen and colleagues report on the novel interaction of SUN1, a component of the linker of nucleoskeleton and cytoskeleton (LINC) complex, with the protein Speedy A (SPDYA). The authors determined the crystal structure of the extended Ringo domain of SPDYA in complex with CDK2 and a short, 20 residue-region of SUN1. The authors used in vivo assays to demonstrate that the SUN1-SPDYA interaction is necessary for gametogenesis in mice.

Overall, this study shows the role of SUN1 and SPDYA in the association of telomeres with the LINC complex and could be a valuable contribution to our understanding of meiotic prophase I progression. The manuscript contains excellent quality in vivo data, however structural part is weak and unconvincing.

Major concerns:

The resolution of the reported here structure, which is 3.2 angstrom, is very low for such a relatively small complex, and is lower than the resolution of the published structure of the SPDYA-CDK2 complex (2.7 angstrom). To be suitable for publication, the SPDYA-CDK2-SUN1 complex structure needs to be re-determined, and the resolution has to be less than 3.0 angstrom, ideally, less than 2.7 angstrom. At the current resolution 3.2, only basic contours of the proteins' chains can be observed.

To strengthen the structural part, binding affinities of WT and mutated SPDYA and SUN1 have to be measured and compared.

Minor: Some IF images shown in Figures 2-6 could be moved to a supplement.

Reviewer #2 (Remarks to the Author):

Chen et al investigate the role of an interaction between SUN1 and SPDYA in gametogenesis. They generate protein structures of these two proteins interacting and identify a Trp amino acid that when mutated to an Arg or Ala disrupts the interaction between these two proteins. When this mutation is introduced homozygously into mice, these mice fail to properly go through gametogenesis. The authors use Hi-C and microscopy analysis to show that the organization of meiotic chromosomes is clearly different as a consequence of this mutation. The cells fail to perform programmed DSB repair and therefore cannot do homologous recombination. The authors also show that the interactions with the nuclear lamina is disrupted.

In general, I believe the data is well presented and interesting. The mutation seems to lead to a spectacular phenotype in the Hi-C map.

I believe the authors can do a more thorough analysis on the Hi-C data though. I was missing the following analyses:

*relative contact probability plots to show the overall decay of the Hi-C signal

- * A/B compartment analysis: does this change between the mutant and the wild-type.
- * saddle plots/compartment strength analysis
- * the WT seems to show stronger interactions in trans, please quantify
- * please zoom into two chromosomes, a single chromosome and to the TAD level.
- * Wang et al 2019 Mol Cell have shown a developmental series for mouse gametogenesis, please compare the Hi-C data to this data, as it may provide a hint into which developmental stage the cells get stuck.

Other points:

- * Figure 1g does not have loading controls. Why is there a difference in the anti-Flag levels between the W150R and W150A?
- * Many IF images lack quantification. Although the qualitative differences are often striking. It would be useful to have the quantitative information as well.

Reviewer #3 (Remarks to the Author):

In the manuscript “The SUN1-SPDYA interaction plays an essential role in meiosis prophase I”, Chen et al. characterize an important protein-protein interaction between the SUN domain protein SUN1 and SPDYA, a regulator of meiotic chromosome structure. In most eukaryotes, chromosome ends are linked to the nuclear envelope in meiotic prophase, where they are clustered and then moved (by molecular motors on the outside of the nuclear envelope) to mediate accurate inter-homolog pairing. Using a two-hybrid screen followed by a more robust localization screen, the authors identified a direct interaction between SUN1 and SPDYA, and solved the crystal structure of a tripartite SUN1-SPDYA-CDK2 complex. Based on the structure, the authors identified the critical residue Trp150 on SUN1. By mutating this residue, they can disrupt the interaction between SUN1 and SPDYA. Finally, they looked into the impact of this mutation in vivo using transgenic mice, and found several interesting phenotypes including problematic chromosome alignment in meiosis prophase I; defective DSB repair and incomplete SC formation in prophase I; loss of telomere-LINC connection and telomere-INM (inner nuclear membrane) attachment; congregation of LINC complex and nuclear lamina; a ring-shaped telomere structure, which they first described in this manuscript, is missing; telomere Phusion at the INM. Overall, they provide convincing data demonstrating the critical interaction between SUN1 and SPDYA for telomere organization as well as proper homologous chromosome alignment during meiosis prophase I. Overall, this manuscript provides important new insights that will have an impact on the meiosis field, and I support publication provided the authors can address the questions and concerns listed below.

Broad questions:

This manuscript provides convincing data on the linkage between the SUN1-SPDYA interaction and various phenotypes, but it lacks causal linkage for those phenotypes. For example, how does SUN1-SPDYA affect the DSB repair, and does it have anything to do with CDK2? Indeed, this interaction seems like it is mostly (entirely?) about specific recruitment of CDK2, suggesting an important role for CDK2 in generating or stabilizing telomere-LINC complex associations. Can the authors address this question more directly, perhaps in the Discussion?

Another example would be LINC complex and lamina dynamics in the mutant mouse strains. Does loss of LINC complex have anything to do with CDK2? Does SPDYA interact with telomere proteins so that SPDYA might act as an adapter for LINC-telomere interactions?

The ring-shaped structure the authors describe for the telomere is very interesting, and the SUN1-SPDYA interaction seems essential for this structure. However, the underlying mechanism seems unclear at this stage. While it's fine to not having a complete functional picture at this point, it would be beneficial to discuss how these structures might be formed. Does the assembly of such a structure occur exclusively on the INM? Is the LINC complex the actual organizer for this ring-shaped structure?

Other questions/suggestions:

The authors' use of different numbering for the critical tryptophan residue in SUN1 (Trp150 or Trp151, depending on human vs. mouse) could be confusing to a reader – indeed I initially thought this was a typo. I suggest clarifying upon first mention that the numbering refers to the human versus mouse sequences.

Figure 2F: The relative Hi-C contact frequency should be on the same scale for the two graphs. Moreover, I suggest that the authors provide more analysis of the Hi-C data to support their statement (lines 157-164) that in the W151R mutant, chromosome ends are not as clustered as in wild-type. At the very least, I suggest replacing the genome-wide Hi-C maps with maps of a single trans chromosome pair (e.g. chromosomes 1 vs. 2, or 3 vs. 4), and marking the centromeric and telomeric regions to clarify this point to a reader. This would make it clearer to a reader how the authors arrived at the graphs shown in Figure 2G.

Figure 5d: Lamin B1 distribution interfered by the loss of LINC-telomere is not supported given that the wild type has a similar congregation phenomenon. The lamina dynamics mechanism is unclear whether in WT or mutant strains. The LINC complex might behave just as an adapter between lamina and telomere, and have no effect on lamina dynamics based on the data shown in Figure 5 and Supplementary Figure 6.

Line 308: The conclusion that the ring-shaped structure provides the basis for telomere-INM attachment is not well established. Yes, it's true that the mutant lost the ring-shaped structure, and it lost telomere-INM attachment partially (some of them are TTM based). But there is no evidence yet supporting the ring-shaped structure is the basis for telomere-INM attachment. It would be good to look at the TTM-based telomere-INM attachment by super-resolution microscopy, if possible.

Reviewer #4 (Remarks to the Author):

In this manuscript, Chen et al. found the direct interaction between SUN1 and SPDYA. The authors then determined the crystal structure of SUN1-CDK2-SPDY complex, and identified the W150 residue in SUN1 that is critical for binding to SPDYA. They also generated the W151R knock-in mutant mice of Sun1, and showed that the mutant mice exhibited the defects in gametogenesis, meiotic DSB repair, the telomere attachment to nuclear envelope, the distribution of LINC complex, and the

telomere structure and integrity. Essentially, this study can contribute to understanding the meiotic telomere structure. However, I have some criticisms shown below, to which the authors need to reply.

1. Generation of mutant mice: The authors identified the W150 residue in SUN1 that is critical for binding to SPDY, and showed that W150R mutation disrupts the interaction with SPDYA. However, they generated the W151R, not W150R, knock-in mutant mice of Sun1. The authors should explain that this is due to the species difference, and the human W150 corresponds to mouse W151.

2. Extended Data Fig. 1d, e: By a loss-of function assay, the 136-155 region of SUN1 was shown to be required for interaction with SPDYA. Can the fusion protein of the 136-155 region with the C-terminal region from the TM domain recruit SPDYA to nuclear membrane like wild-type SUN1? The authors succeeded the crystallization of the SUN1(131-160)-SPDYA-CDK2, indicating that the 131-160 region of SUN1 can interact with SPDYA. However, it was not shown that the interaction via this small region is almost the same as that via wild-type SUN1.

3. Fig. 4c: The W151R zygotene and pachytene-like spermatocytes exhibited reduced intensity of SPDYA at telomeres. However, a showing the whole cell image is not enough to conclude this. The authors should quantify the SPDYA fluorescence intensity by using the fluorescence-negative region as a control. The authors also should explain or speculate why the amount of SPDYA at telomeres reduces in the W151R zygotene and pachytene-like spermatocytes.

4. Fig. 4d, e: About half of telomeres in W151R mutant nuclei remained at the inner nuclear membrane (INM). Since the SUN1-SPDYA interaction is critical for the connection of telomere with the INM, one would expect that none of the telomeres, not a half, remained at the INM. The authors should explain or speculate why half of telomeres still remained at the INM. Does similar amount of telomeres (about half) in Sun1-deficient mutant nuclei remained at the INM?

Point-to-point responses to reviewers' comments

The SUN1-SPDYA interaction plays an essential role in meiosis prophase I

NCOMMS-20-26947-T

Reviewer #1:

In this manuscript, Chen and colleagues report on the novel interaction of SUN1, a component of the linker of nucleoskeleton and cytoskeleton (LINC) complex, with the protein Speedy A (SPDYA). The authors determined the crystal structure of the extended Ringo domain of SPDYA in complex with CDK2 and a short, 20 residue-region of SUN1. The authors used in vivo assays to demonstrate that the SUN1-SPDYA interaction is necessary for gametogenesis in mice.

Overall, this study shows the role of SUN1 and SPDYA in the association of telomeres with the LINC complex and could be a valuable contribution to our understanding of meiotic prophase I progression. The manuscript contains excellent quality in vivo data, however structural part is weak and unconvincing.

Major concerns:

1- The resolution of the reported here structure, which is 3.2 angstrom, is very low for such a relatively small complex, and is lower than the resolution of the published structure of the SPDYA-CDK2 complex (2.7 angstrom). To be suitable for publication, the SPDYA-CDK2-SUN1 complex structure needs to be re-determined, and the resolution has to be less than 3.0 angstrom, ideally, less than 2.7 angstrom. At the current resolution 3.2, only basic contours of the proteins' chains can be observed.

We thank the reviewer for pointing out this issue. During the past three months, we have tried everything we can to improve the crystal structure quality, including optimizing crystallization and cryo-protectant conditions, testing crystal dehydration, trying different annealing time, etc. Unfortunately, these efforts did not improve the crystal diffraction quality. In addition, we have also tried many different strategies to improve the refinement of the structure, including rigid-body fitting, conjugate gradient minimization, unrestrained B-factor refinement, individual B-factor refinement, TLS anisotropic displacements, simulated annealing, with different programs such as CCP4, Refmac, Phenix and Rosetta. We have improved the refinement statistics of the structure (R_{free} is now 30.0% in

the revised structure compared with 31.2% in the original manuscript). In the revised Supplementary Table 1, we have included highest resolution shell and have removed all Ramachandran outliers. We are confident that the electron density map at the interface between SUN1 and SPDYA is good enough to generate the correct structure information, which is consistent with our mutagenesis data of co-IP (revised Fig. 1g) and membrane recruitment assays (revised Supplementary Fig. 3a). In addition, following the reviewer’s suggestion, we have also measured the binding affinities of WT and mutant SUN1-SPDYA interaction using surface plasmon resonance analysis (revised Supplementary Fig. 3b). Mutations at the interface completely disrupted the SUN1- SPDYA interaction, further confirming the correctness of the structure information.

2- To strengthen the structural part, binding affinities of WT and mutated SPDYA and SUN1 have to be measured and compared.

We thank the reviewer for raising this good point. We have compared the binding affinities of WT and mutated SUN1₁₃₁₋₁₆₀ fragment with the SPDYA_{ERD}-CDK2 complex *in vitro* using surface plasmon resonance experiments (**Reviewer Fig. 1**). The results show that WT SUN1₁₃₁₋₁₆₀ but not W151R or W151A mutant SUN1₁₃₁₋₁₆₀ binds to the SPDYA_{ERD}-CDK2 complex in a concentration-dependent manner. The estimated K_D for the interaction between SUN1₁₃₁₋₁₆₀ and SPDYA_{ERD}-CDK2 is about 30.1 μ M.

Reviewer Fig. 1 (revised Supplementary Fig. 3b): Surface plasmon resonance measurements show that WT SUN1₁₃₁₋₁₆₀ binds to the SPDYA_{ERD}-CDK2 complex in a concentration-dependent manner with a K_D of 30.1 μ M.

Minor:

3- Some IF images shown in Figures 2-6 could be moved to a supplement.

We have moved some IF images from main Figures to Supplementary Figures in the revised manuscript as listed below.

Original	Revised
Fig. 2e	Supplementary Fig. 5c
Fig. 3a-f	Supplementary Fig. 7a-f
Fig. 4b, f, g	Supplementary Fig. 8a, d, e

Reviewer #2:

Chen et al investigate the role of an interaction between SUN1 and SPDYA in gametogenesis. They generate protein structures of these two proteins interacting and identify a Trp amino acid that when mutated to an Arg or Ala disrupts the interaction between these two proteins. When this mutation is introduced homozygously into mice, these mice fail to properly go through gametogenesis. The authors use Hi-C and microscopy analysis to show that the organization of meiotic chromosomes is clearly different as a consequence of this mutation. The cells fail to perform programmed DSB repair and therefore cannot do homologous recombination. The authors also show that the interactions with the nuclear lamina is disrupted.

In general, I believe the data is well presented and interesting. The mutation seems to lead to a spectacular phenotype in the Hi-C map.

4- I believe the authors can do a more thorough analysis on the Hi-C data though. I was missing the following analyses:

*relative contact probability plots to show the overall decay of the Hi-C signal

*A/B compartment analysis: does this change between the mutant and the wild-type.

* saddle plots/compartment strength analysis

*the WT seems to show stronger interactions in trans, please quantify

* please zoom into two chromosomes, a single chromosome and to the TAD level.

* Wang et al 2019 Mol Cell have shown a developmental series for mouse gametogenesis, please

compare the Hi-C data to this data, as it may provide a hint into which developmental stage the cells get stuck.

We thank the reviewer for this constructive comment. We have added more Hi-C analyses in the revised manuscript.

(1) We added contact probability plots ($P(s)$) in **Reviewer Fig. 2a** to illustrate the decay of chromatin contact probability as a function of genome distances. *W151R* spermatocytes exhibit a power-law decay with a slope of ~ 0.6 between 100 kb and 1 Mb, comparable to previously published meiotic Hi-C datasets.

(2) We performed A/B compartment analysis and included eigen1 profiles for chromosome 5 in **Reviewer Fig. 2b**. Hi-C heatmaps and eigen1 profiles indicate that whether a genomic region belongs to compartment A or B is not significantly affected by the *W151R* mutation.

(3) We further quantified the compartment strength in *W151R* spermatocytes using saddle plots in **Reviewer Fig. 2c** and **2d** as the reviewer suggested. These analyses indicated that the compartment strength in *W151R* spermatocytes is moderately weakened compared to WT zygotene spermatocytes.

(4) The *trans* interaction heatmaps for WT spermatocytes exhibited an 'X' shape pattern with most signals enriched near the chromosome ends, consistent with cytological observations of the coalescence of telomeres. On the other hand, the *trans* interactions for *W151R* spermatocytes appeared more dispersed, leading to an impression of decreased total *trans* interactions in the mutant. We quantified the ratio of *cis* interactions versus the total chromatin interactions for our mutant dataset and found that the *W151R* dataset exhibits a *cis*/total ratio of $\sim 60\%$ (**Reviewer table 1**). In contrast, the *cis*/total ratios for previously published WT zygotene and pachytene datasets are from 80% to 89% (Wang_P, Patel_P, and Patel_Z in **Reviewer table 1**) [1, 2]. However, we noted that the choices of restriction enzymes are different in our study and previous studies (DpnII for our dataset and MboI for Wang et al. and Patel et al. datasets) [1, 2]. While this manuscript was under review, we conducted a parallel study to investigate the progressive chromosome organization changes during the meiotic prophase and generated more WT Hi-C datasets using the DpnII enzyme (Zuo_L, Zuo_Z, Zuo_P, and Zuo_D in **Reviewer table 1**). A manuscript describing these unpublished data is currently also under review at Nature Communications. We found that these newly generated WT datasets using DpnII all exhibit $\sim 60\%$ *cis* ratios, which are similar to the *W151R* dataset. We reasoned that the differences in *cis* pair ratio are most likely caused by technical variations, rather than reflecting biological differences in WT and mutant spermatocytes. Therefore, we did not emphasize the changes in total *trans* interactions in the revised manuscript. Instead, we normalized the observed Hi-C interactions with the expected values in all heatmaps in the manuscript (**Reviewer Fig. 2** and **Reviewer Fig. 3**) to minimize the technical variations between the *W151R* dataset and the published datasets used for comparison.

(5) We have modified **Reviewer Fig. 3a** (original Fig. 2f) to include the zoomed-in views of a 4 Mb region in addition to the entire chromosome 2.

(6) As described above, we compared the *W151R* spermatocytes with the published datasets from both the Patel *et al.* and the Wang *et al.* studies [1, 2]. Although the $P(s)$ plot of the mutant spermatocytes exhibited a shape that is not identical to either WT zygotene or WT pachytene spermatocytes, it is more similar to WT zygotene. This Hi-C data is consistent with the IF staining result of Synaptonemal Complex components (revised Fig. 2f), which indicates a meiotic arrest at zygotene stage in *W151R* testes.

Reviewer Fig. 2 (revised Supplementary Fig. 6): Disruption of the SUN1-SPDYA interaction affects 3D genome organization. **a**, P(s) curves indicate relationships between chromatin contact probability and genomic separation for chromatin interactions on autosomes (left) and the X chromosome (right) in WT zygote and pachytene spermatocytes as well as *W151R* mutant spermatocytes. **b**, Heatmaps binned at 50 kb resolution showing normalized Hi-C interactions (observed/ expected) for Chr5. **c**, Saddle plots indicate the extent of genome compartmentalization in each dataset. **d**, Saddle strength profiles quantify the strength of genome compartmentalization by calculating ratios of cumulative corner interaction scores from the saddle plots ((AA+BB)/ (AB+BA)).

Reviewer Fig. 3 (revised Fig. 3): Disruption of the SUN1-SPDYA interaction impairs extensive alignment of chromosome ends. **a**, Heatmaps showing normalized Hi-C interactions (observed/expected) for the entire Chr2 (top, 50-kb bin) or a 4 Mb Chr2 region (bottom, 10-kb bin) in WT zygotene and *W151R* pachytene-like spermatocytes. **b**, Heatmaps showing normalized Hi-C interactions between Chr1 and Chr2 (50-kb bin) in WT zygotene and *W151R* pachytene-like spermatocytes. **c**, Boxplot shows changes in normalized inter-chromosomal interactions among 10 Mb regions at the centromeric (Cen) or non-centromeric (Tel) ends of chromosomes in *W151R* mutant.

	cis Pairs	Unique Valid Pairs	% cis Pairs
Wang_P	241,592,148	301,040,000	80.25%
Patel_P	432,990,224	487,112,863	88.89%
Patel_Z	298,659,261	351,515,442	84.96%
Zuo_L	160,513,369	263,447,520	60.93%
Zuo_Z	237,700,676	381,769,629	62.26%
Zuo_P	269,739,994	480,580,864	56.13%
Zuo_D	425,596,934	694,334,242	61.30%
W151R_mutant	116,600,701	191,374,498	60.93%

Reviewer table 1: Ratio of *cis* interactions versus the total chromatin interactions in different Hi-C datasets.

Other points:

5- Figure 1g does not have loading controls. Why is there a difference in the anti-Flag levels between the W150R and W150A?

Thank you very much for pointing out this issue. We have redone the co-IP experiment, obtained better images that give the same conclusion – the presence of CDK2 substantially enhances the interaction between SUN1 and SPDYA and mutations of SUN1 completely abolish this interaction (**Reviewer Fig. 4**). In this experiment, anti-GAPDH was used as a loading control in Western blot analysis.

Reviewer Fig. 4 (revised Fig. 1g): Co-IP analysis of ectopically expressed human Flag-SUN1₁₋₂₁₀, Myc-SPDYA and CDK2 in HEK293T cells.

6- Many IF images lack quantification. Although the qualitative differences are often striking. It would be useful to have the quantitative information as well.

We thank the reviewer for this suggestion. We have quantified more IF images and included new quantification data in the revised manuscript as listed below.

Revised Figures	Quantification data
Fig. 2c	Fig. 2d
Fig. 2f	Fig. 2g
Fig. 4b	Fig. 4c
Supplementary Fig. 5c	Supplementary Fig. 5d
Supplementary Fig. 8e	Supplementary Fig. 8f
Supplementary Fig. 9a	Supplementary Fig. 9b
Supplementary Fig. 9c and d	Supplementary Fig. 9e

Reviewer #3

In the manuscript “The SUN1-SPDYA interaction plays an essential role in meiosis prophase I”, Chen et al. characterize an important protein-protein interaction between the SUN domain protein SUN1 and SPDYA, a regulator of meiotic chromosome structure. In most eukaryotes, chromosome ends are linked to the nuclear envelope in meiotic prophase, where they are clustered and then moved (by molecular motors on the outside of the nuclear envelope) to mediate accurate inter-homolog pairing. Using a two-hybrid screen followed by a more robust localization screen, the authors identified a direct interaction between SUN1 and SPDYA, and solved the crystal structure of a tripartite SUN1-SPDYA-CDK2 complex. Based on the structure, the authors identified the critical residue Trp150 on SUN1. By

mutating this residue, they can disrupt the interaction between SUN1 and SPDYA. Finally, they looked into the impact of this mutation in vivo using transgenic mice, and found several interesting phenotypes including problematic chromosome alignment in meiosis prophase I; defective DSB repair and incomplete SC formation in prophase I; loss of telomere-LINC connection and telomere-INM (inner nuclear membrane) attachment; congregation of LINC complex and nuclear lamina; a ring-shaped telomere structure, which they first described in this manuscript, is missing; telomere Phusion at the INM. Overall, they provide convincing data demonstrating the critical interaction between SUN1 and SPDYA for telomere organization as well as proper homologous chromosome alignment during meiosis prophase I. Overall, this manuscript provides important new insights that will have an impact on the meiosis field, and I support publication provided the authors can address the questions and concerns listed below.

Broad questions:

7- This manuscript provides convincing data on the linkage between the SUN1-SPDYA interaction and various phenotypes, but it lacks causal linkage for those phenotypes. For example, how does SUN1-SPDYA affect the DSB repair, and does it have anything to do with CDK2? Indeed, this interaction seems like it is mostly (entirely?) about specific recruitment of CDK2, suggesting an important role for CDK2 in generating or stabilizing telomere-LINC complex associations. Can the authors address this question more directly, perhaps in the Discussion?

We thank the reviewer for the insightful comment. Meiotic DSB repair correlates with SC formation and homologous recombination and may behave as an indicator for homologous pairing process [3]. The timing of meiotic DSB repair (mainly carried out at zygotene stage) is after telomere attachment to the INM, when CDK2 is already located at the telomere regions together with SPDYA. We think that CDK2 is not directly implicated in autosome DSB repair. Our *in-vitro* experiments showed that CDK2 helps the SUN1-SPDYA interaction (revised Fig. 1g), which may promote the telomere-LINC complex connection to facilitate homolog search and pairing, and may ultimately affect DSB repair.

We are happy that the reviewer's comment is consistent with our conclusion. The *in-vitro* assays in our study show that CDK2 enhances the SUN1-SPDYA interaction, suggestive of an important role for CDK2 in establishing or stabilizing the telomere-LINC connection. To clarify this point, we added one paragraph in the discussion to address this issue (page 19, lines 417-424).

“CDK2 is a serine/threonine protein kinase that plays an important role in mitotic cell cycle regulation [4]. Deletion of CDK2 leads to meiotic arrest at a pachytene-like stage, suggesting that CDK2 also play an essential role in meiosis [5, 6]. In meiotic prophase I, CDK2 is found at telomeres on the INM from leptotema towards diplotene and in the XY bodies at pachytene stage [7]. Our data reveal that CDK2 can enhance the SUN1-SPDYA interaction and facilitate the formation of the meiosis-specific telomere-LINC supramolecular complex. Whether CDK2's kinase activity plays a direct role in the meiotic prophase is still not clear and awaits future investigations.”

8- Another example would be LINC complex and lamina dynamics in the mutant mouse strains. Does loss of LINC complex have anything to do with CDK2? Does SPDYA interact with telomere proteins so that SPDYA might act as an adapter for LINC-telomere interactions?

We thank the reviewer for pointing out this issue. A previous study has revealed a direct interaction between SPDYA and TRF1 [8], so that SPDYA might act as an adapter for LINC-telomere interactions.

CDK2 enhances SUN1-SPDYA interaction which is essential to the telomere-LINC complex connection. So we believe that CDK2 also contributes to the LINC-telomere interactions. We have added one paragraph in the discussion to address this issue as following (page 19, lines 417-424).

“CDK2 is a serine/threonine protein kinase that plays an important role in mitotic cell cycle regulation [4]. Deletion of CDK2 leads to meiotic arrest at a pachytene-like stage, suggesting that CDK2 also play an essential role in meiosis [5, 6]. In meiotic prophase I, CDK2 is found at telomeres on the INM from leptotema towards diplotene and in the XY bodies at pachytene stage [7]. Our data reveal that CDK2 can enhance the SUN1-SPDYA interaction and facilitate the formation of the meiosis-specific telomere-LINC supramolecular complex. Whether CDK2’s kinase activity plays a direct role in the meiotic prophase is still not clear and awaits future investigations.”

9- The ring-shaped structure the authors describe for the telomere is very interesting, and the SUN1-SPDYA interaction seems essential for this structure. However, the underlying mechanism seems unclear at this stage. While it’s fine to not having a complete functional picture at this point, it would be beneficial to discuss how these structures might be formed. Does the assembly of such a structure occur exclusively on the INM? Is the LINC complex the actual organizer for this ring-shaped structure?

We thank the reviewer for this insightful comment. We speculate that the telomere ring-shaped structure is a consequence of the telomere-LINC connection and telomere movements along the INM. We have added more details in the discussion about how this ring-shaped structure is formed (page 18, lines 379-395).

“First, interaction between TERB1 of the TTM complex and TRF1 in the telomere-associated shelterin complex initiates the connection of telomeres with the INM in leptotene spermatocytes (Fig. 7e, stage 1)[9, 10]. At this stage, most telomeres adopt linear conformations and likely not all the telomere-associated shelterin complexes bind to the TTM complexes (Fig. 7e, stage 1). Loss of this TTM-mediated telomere-INM interaction completely prevents telomere attachment to the INM[11, 12]. Next, the LINC complex binds to the TTM-tethered telomeres promoted by the direct interaction between SUN1 and SPDYA to achieve a more stable association of telomeres with the INM (Fig. 7e, stage 2). Then, dynein-dependent movement of telomeres at the INM transduced through the LINC complex allows telomeres to encounter more TTM and LINC complexes. Finally, all the telomere associated

shelterin complexes are saturated by both the TTM and LINC complexes so that the entire telomere region is tightly attached to the INM to form in a ring-shaped supramolecular architecture (Fig. 7e, stages 3 and 4). It is likely that the LINC complex is a key organizer for this supramolecular arrangement. Disruption of the SUN1-SPDYA interaction by the *Sun1*^{W151R} mutation abolishes the telomere-LINC complex connection and causes a complete loss of telomeric ring-sharp architecture (Figs. 4a and 6a), resulting in partial telomere detachment from the INM and reduced stability of SPDYA at telomeres (Fig. 4b-e).”

We believe that the telomere ring-shaped architecture occurs exclusively on the INM, because these structures are likely a consequence of telomere-LINC complex connection and telomere movements along the INM, which depend on the nuclear envelope-spanning LINC complex for force transmission.

We also believe that the LINC complex is a key organizer for this ring-shaped structure. LINC complex-dependent telomere movements allow to recruit more LINC and TTM complexes to saturate the telomere-bound shelterin complex so that the entire telomere region is attached to the INM to form the ring-shaped structure.

10- The authors’ use of different numbering for the critical tryptophan residue in SUN1 (Trp150 or Trp151, depending on human vs. mouse) could be confusing to a reader – indeed I initially thought this was a typo. I suggest clarifying upon first mention that the numbering refers to the human versus mouse sequences.

We thank the reviewer for pointing out this issue. We have modified the text as the following (page 6, lines 124-127).

“The same conclusion was also obtained when the interactions between mouse Sun1 mutants (mutations at Trp151, equivalent to human SUN1^{Trp150}) and Spdya-Cdk2 were analyzed, consistent with the fact that the interface residues are highly conserved between human and mouse proteins (Supplementary Fig. 3c-e).”

11- Figure 2F: The relative Hi-C contact frequency should be on the same scale for the two graphs. Moreover, I suggest that the authors provide more analysis of the Hi-C data to support their statement (lines 157-164) that in the W151R mutant, chromosome ends are not as clustered as in wild-type. At the very least, I suggest replacing the genome-wide Hi-C maps with maps of a single trans chromosome pair (e.g. chromosomes 1 vs. 2, or 3 vs. 4), and marking the centromeric and telomeric regions to clarify this point to a reader. This would make it clearer to a reader how the authors arrived at the graphs shown in Figure 2G.

We thank the reviewer for this suggestion. In the revised manuscript, we have modified **Reviewer Fig. 5** (to replace the original Fig. 2F) to only show normalized *trans* interactions (observed

interactions/expected) between chromosomes 1 and 2. The heatmaps for WT spermatocytes and *W151R* mutant spermatocytes are now shown on the same scale. Centromeric regions are indicated in the heatmaps for a clearer presentation.

We have also added more Hi-C analyses in the revised text, including (1) contact probability plots (P(s)) to illustrate the decay of chromatin contact probability as a function of genome distances (**Reviewer Fig. 2a**), (2) A/B compartment analysis for chromosome 5 to show that *W151R* mutation does not significantly affect A/B compartment distribution (**Reviewer Fig. 2b**), (3) quantification of the compartment strength to show that the compartment strength in *W151R* spermatocytes was moderately weakened (**Reviewer Fig. 2c and 2d**).

Reviewer Fig. 5 (revised Fig. 3b): Heatmaps showing normalized Hi-C interactions between Chr1 and Chr2 (50-kb bin) in WT zygotene and *W151R* pachytene-like spermatocytes.

12- Figure 5d: Lamin B1 distribution interfered by the loss of LINC-telomere is not supported given that the wild type has a similar congregation phenomenon. The lamina dynamics mechanism is unclear whether in WT or mutant strains. The LINC complex might behave just as an adapter between lamina and telomere, and have no effect on lamina dynamics based on the data shown in Figure 5 and Supplementary Figure 6.

Thanks for pointing out this issue. So far the mechanism for lamina dynamics is unclear. In both *W151R* and *Sun1*^{-/-} spermatocytes, the lamina exhibits a similar congregation phenomenon as in WT spermatocytes (revised Fig. 5c and supplementary Fig. 9c-g). However, the signal(s) that triggers the dispersion (a part of dynamics) of the congregated lamina is missing, resulting in much higher population of mutant spermatocytes showing lamina congregation (revised Fig.5c and supplementary Fig. 9c-g). This observation suggested that lamina dynamics is affected by the loss of LINC-telomere connection.

To clearly specify this point, we have modified the text as following (page 14, lines 292-294).

“Although the mechanism for lamina dynamics is still not clear, the dispersion of congregated lamina

being affected in both *W151R* and *Sun1*^{-/-} spermatocytes suggests that loss of the LINC-telomere connection interferes with lamina dynamics during meiotic prophase.”

13- Line 308: The conclusion that the ring-shaped structure provides the basis for telomere-INM attachment is not well established. Yes, it’s true that the mutant lost the ring-shaped structure, and it lost telomere-INM attachment partially (some of them are TTM based). But there is no evidence yet supporting the ring-shaped structure is the basis for telomere-INM attachment. It would be good to look at the TTM-based telomere-INM attachment by super-resolution microscopy, if possible.

We thank the reviewer for this point. We have modified the text as following (page 16, lines 339-341).

“In aggregate, we conclude that the SUN1-SPDYA interaction is essential for the assembly of the telomeric ring-shaped architecture at the INM, which is independent of homolog synapsis.”

We also examined the TTM-based telomere-INM attachment (telomeres with MAJIN foci) in *W151R* spermatocytes as suggested and observed no ring structures for telomeres or MAJIN (**Reviewer Fig. 6** and **Reviewer Fig. 7**), suggestive of an essential role of the SUN1-SPDYA interaction in telomeric ring structure assembly.

Reviewer Fig. 6 (revised Fig. 6d, top): IF analysis by STED of spermatocyte spreads stained for SYCP3 (green) and MAJIN (red), TERB2 (red).

Reviewer Fig. 7 (revised Supplementary Fig. 10d): IF analysis by STED of spermatocyte spreads stained for TRF1 (red) and MAJIN (cyan).

Reviewer #4

In this manuscript, Chen et al. found the direct interaction between SUN1 and SPDYA. The authors then determined the crystal structure of SUN1-CDK2-SPDY complex, and identified the W150 residue in SUN1 that is critical for binding to SPDYA. They also generated the W151R knock-in mutant mice of Sun1, and showed that the mutant mice exhibited the defects in gametogenesis, meiotic DSB repair, the telomere attachment to nuclear envelope, the distribution of LINC complex, and the telomere structure and integrity. Essentially, this study can contribute to understanding the meiotic telomere structure. However, I have some criticisms shown below, to which the authors need to reply.

14- Generation of mutant mice: The authors identified the W150 residue in SUN1 that is critical for binding to SPDY, and showed that W150R mutation disrupts the interaction with SPDYA. However, they generated the W151R, not W150R, knock-in mutant mice of Sun1. The authors should explain that this is due to the species difference, and the human W150 corresponds to mouse W151.

We thank the reviewer for this point. We have clarified this point in the revised manuscript as the following (page 6, lines 124-127).

“The same conclusion was also obtained when the interactions between mouse Sun1 mutants (mutations at Trp151, equivalent to human SUN1^{Trp150}) and Spdya-Cdk2 were analyzed, consistent with the fact that the interface residues are highly conserved between human and mouse proteins (Supplementary Fig. 3c-e).”

15- Extended Data Fig. 1d, e: By a loss-of function assay, the 136-155 region of SUN1 was shown to be required for interaction with SPDYA. Can the fusion protein of the 136-155 region with the C-terminal region from the TM domain recruit SPDYA to nuclear membrane like wild-type SUN1? The authors succeeded the crystallization of the SUN1(131-160)-SPDYA-CDK2, indicating that the 131-160 region of SUN1 can interact with SPDYA. However, it was not shown that the interaction via this small region is almost the same as that via wild-type SUN1.

We thank the reviewer for this advice. To address this point, we fused the SUN1₁₃₆₋₁₅₅ segment with membrane-bound MAJIN and ectopically co-expressed the fusion protein with SPDYA in U-2 OS cells. SUN1₁₃₆₋₁₅₅-MAJIN but not MAJIN could efficiently recruit SPDYA to the INM (**Reviewer Fig. 8**), indicating that residues 136-155 of SUN1 are sufficient to recruit SPDYA to nuclear membrane.

We didn't fuse SUN1₁₃₆₋₁₅₅ with the C-terminal TM domain of SUN1, because the C-terminal region of SUN1 by itself only exhibits poor INM-localization ability [13].

Reviewer Fig. 8 (revised Supplementary Fig. 1f): Equator images of U-2 OS cells expressing MAJIN^{DB}-GFP or SUN1₁₃₆₋₁₅₅-MAJIN^{DB}-GFP (green) together with Myc-SPDYA (red). The DNA-binding defective mutant of MAJIN (MAJIN^{DB}) exhibits apparent nuclear membrane localization when ectopically expressed in U-2 OS cells [11].

16- Fig. 4c: The W151R zygotene and pachytene-like spermatocytes exhibited reduced intensity of SPDYA at telomeres. However, a showing the whole cell image is not enough to conclude this. The authors should quantify the SPDYA fluorescence intensity by using the fluorescence-negative region as a control. The authors also should explain or speculate why the amount of SPDYA at telomeres reduces in the W151R zygotene and pachytene-like spermatocytes.

We have quantified the SPDYA fluorescence intensity at telomeres in *W151R* zygotene and pachytene-like spermatocytes (**Reviewer Fig. 9**). The result clearly shows that *W151R* zygotene and pachytene-like spermatocytes exhibited reduced intensity of SPDYA staining at telomeres.

It is likely that the SUN1-SPDYA interaction promotes the telomere-LINC complex connection to establish a supramolecular telomere complex at the INM, which in turn stabilizes the localization of SPDYA at telomeres. Conversely, loss of the telomere-LINC connection reduces the amount of SPDYA at telomeres in *W151R* zygotene and pachytene-like spermatocytes.

To emphasize this point, we modified the text in the discussion (page 18, lines 391-395).

“It is likely that the LINC complex is a key organizer for this supramolecular arrangement. Disruption of the SUN1-SPDYA interaction by the *Sun1*^{W151R} mutation abolishes the telomere-LINC complex connection and causes a complete loss of telomeric ring-sharp architecture (Figs. 4a and 6a), resulting in partial telomere detachment from the INM and reduced stability of SPDYA at telomeres (Fig. 4b-e).”

Reviewer Fig. 9 (revised Fig. 4c): Quantification of the relative intensity of SPDYA foci at telomeres.

17- Fig. 4d, e: About half of telomeres in W151R mutant nuclei remained at the inner nuclear membrane (INM). Since the SUN1-SPDYA interaction is critical for the connection of telomere with the INM, one would expect that none of the telomeres, not a half, remained at the INM. The authors should explain or speculate why half of telomeres still remained at the INM. Does similar amount of telomeres (about half) in Sun1-deficient mutant nuclei remained at the INM?

We thank the reviewer for raising this point. SUN1 and MAJIN are two membrane proteins that provide binding sites for telomere attachment at the INM. Previous studies have revealed the structural basis for telomere attachment to the INM via the TERB1-TERB2-MAJIN (TTM) complex [9, 11, 12, 14]. Disruption of the TTM complex completely depletes meiotic telomeres from the INM, while knockout of SUN1 only causes partial detachment of telomeres from the INM [11, 12], suggesting that the TTM complex guides the initial telomere-INM interaction while the LINC complex allows the establishment of a supramolecular telomere complex tightly attached to the INM. In this study, we showed that the SUN1-SPDYA interaction was essential for the SUN1-telomere association (revised Fig. 4a). However, the telomere-TTM interaction was partially retained in *W151R* mutant nuclei (**Reviewer Fig. 10b-d**), which resulted in part of telomeres still remain at the INM (**Reviewer Fig. 10e and 10f**). In fact, SUN1 knockout also only caused the detachment of about half telomeres from the INM as revealed in previous studies [11, 12].

To clearly specify this point, we have modified the text as following (page 11, lines 239-247).

“IF-FISH revealed that about half of telomeres in mutant nuclei remained at the INM (Fig. 4d, e). Notably, the same phenomenon was also observed in *Sun1*^{-/-} spermatocytes [11, 12], indicating that the SUN1-SPDYA interaction plays a key role in telomere-INM connection. Many TTM complex foci appeared at the chromosome tips in mutant spreads and IF-FISH analysis revealed a complete co-localization of MAJIN with the INM-located telomeres in *W151R* spermatocytes (Supplementary Fig. 8b-f), suggesting that the residual telomere-INM attachment in *W151R* nuclei was likely mediated by the INM-anchored TTM complex. This result is consistent with our previous data that disruption of the TTM complex completely removes meiotic telomeres from the INM [12].”

Reviewer Fig. 10 (revised Supplementary Fig. 8): Telomeres are partially retained at the nuclear periphery with TTM in mutant spermatocytes. **a-d**, IF staining of spermatocyte chromosome spreads for SYCP3 (green) and KASH5 (red), TERB1 (red), TERB2 (red), or MAJIN (red) as indicated. DNA was stained by DAPI (blue). **e**, Equator images of structurally preserved zygote spermatocytes stained for SYCP3 (green), MAJIN (red) and telomere FISH (TeIC, magenta). **f**, Population of telomeres displaying MAJIN foci in each spermatocyte nucleus as shown in **e**.

Reference

1. Patel, L., et al., *Dynamic reorganization of the genome shapes the recombination landscape in meiotic prophase*. Nat Struct Mol Biol, 2019. **26**(3): p. 164-174.
2. Wang, Y., et al., *Reprogramming of Meiotic Chromatin Architecture during Spermatogenesis*. Mol Cell, 2019. **73**(3): p. 547-561 e6.

3. Hunter, N., et al., *Gamma-H2AX illuminates meiosis*. Nat Genet, 2001. **27**(3): p. 236-8.
4. Morgan, D.O., *Cyclin-dependent kinases: engines, clocks, and microprocessors*. Annu Rev Cell Dev Biol, 1997. **13**: p. 261-91.
5. Viera, A., et al., *CDK2 regulates nuclear envelope protein dynamics and telomere attachment in mouse meiotic prophase*. J Cell Sci, 2015. **128**(1): p. 88-99.
6. Viera, A., et al., *CDK2 is required for proper homologous pairing, recombination and sex-body formation during male mouse meiosis*. J Cell Sci, 2009. **122**(Pt 12): p. 2149-59.
7. Ashley, T., D. Walpita, and D.G. de Rooij, *Localization of two mammalian cyclin dependent kinases during mammalian meiosis*. J Cell Sci, 2001. **114**(Pt 4): p. 685-93.
8. Wang, L., et al., *Dual roles of TRF1 in tethering telomeres to the nuclear envelope and protecting them from fusion during meiosis*. Cell Death Differ, 2018: p. doi:10.1038/s41418-017-0037-8.
9. Shibuya, H., K. Ishiguro, and Y. Watanabe, *The TRF1-binding protein TERB1 promotes chromosome movement and telomere rigidity in meiosis*. Nat Cell Biol, 2014. **16**(2): p. 145-56.
10. Long, J., et al., *Telomeric TERB1-TRF1 interaction is crucial for male meiosis*. Nat Struct Mol Biol, 2017. **24**(12): p. 1073-1080.
11. Shibuya, H., et al., *MAJIN Links Telomeric DNA to the Nuclear Membrane by Exchanging Telomere Cap*. Cell, 2015. **163**(5): p. 1252-66.
12. Wang, Y., et al., *The meiotic TERB1-TERB2-MAJIN complex tethers telomeres to the nuclear envelope*. Nat Commun, 2019. **10**(1): p. 564.
13. Haque, F., et al., *SUN1 interacts with nuclear lamin A and cytoplasmic nesprins to provide a physical connection between the nuclear lamina and the cytoskeleton*. Mol Cell Biol, 2006. **26**(10): p. 3738-51.
14. Duncce, J.M., et al., *Structural basis of meiotic telomere attachment to the nuclear envelope by MAJIN-TERB2-TERB1*. Nat Commun, 2018. **9**(1): p. 5355.

REVIEWERS' COMMENTS

Reviewer #1 (Remarks to the Author):

The authors have adequately addressed my previous comments.

Reviewer #2 (Remarks to the Author):

My main concern was the lack in quantification of the Hi-C data. This has been adequately addressed. I have no further comments.

Reviewer #3 (Remarks to the Author):

Having read through the rebuttal comments and the revised manuscript, I think the authors have satisfactorily answered and addressed my concerns, and indeed those of my co-reviewers. In particular I am convinced that the SUN1-SPDYA interaction is essential for the assembly of the ring-shape structure for the telomere-INM attachment. Apparently, they have done thorough Hi-C analysis in the revision, and clearly showed that the inter-chromosome interactions decrease dramatically in the SUN1 mutant. Whilst it would have been of further interest to the readership to include some data about role of CDK2 kinase activity toward the SUN1-SPDYA interaction and ring-shape structure assembly, I tend to agree with the authors that in this manuscript CDK2 increased the affinity between SUN1-SPDYA and the kinase part could be left to future studies. I therefore now think that the manuscript is acceptable for publication.

Reviewer #4 (Remarks to the Author):

I have carefully read the revised manuscript. All of my comments have been appropriately replied. So, I recommend to publish this revised manuscript in Nature Communication.